# 🦜 PARROT: MULTILINGUAL VISUAL INSTRUCTION TUNING

## ABSTRACT

The rapid development of Multimodal Large Language Models (MLLMs) like GPT-4V has marked a significant step towards artificial general intelligence. Existing methods mainly focus on aligning vision encoders with LLMs through supervised fine-tuning (SFT) to endow LLMs with multimodal abilities, making MLLMs' inherent ability to react to *multiple languages* progressively deteriorate as the training process evolves. We empirically find that the imbalanced SFT datasets, primarily composed of English-centric image-text pairs, lead to significantly reduced performance in non-English languages. This is due to the failure of aligning the vision encoder and LLM with multilingual tokens during the SFT process. In this paper, we introduce PARROT, a novel method that utilizes textual guidance to drive visual token alignment at the language level. PARROT makes the visual tokens condition on diverse language inputs and uses Mixture-of-Experts (MoE) to promote the alignment of multilingual tokens. Specifically, to enhance non-English visual tokens alignment, we compute the cross-attention using the initial visual features and textual embeddings, the result of which is then fed into the MoE router to select the most relevant experts. The selected experts subsequently convert the initial visual tokens into language-specific visual tokens. Moreover, considering the current lack of benchmarks for evaluating multilingual capabilities within the field, we collect and make available a Massive Multilingual Multimodal Benchmark which includes 6 languages, 15 categories, and 12,000 questions, named as **MMMB**. Our method not only demonstrates state-of-the-art performance on multilingual MMBench and MMMB, but also excels across a broad range of multimodal tasks.

## 1 INTRODUCTION

The rapid development of Large Language Models (LLMs), such as GPT-4 (Radford et al., 2018; Brown et al., 2020; OpenAI, 2023a; 2024), has gained significant attention. However, LLMs are limited to processing a single textual modality. The expansion into visual modalities has endowed LLMs with multimodal capabilities (Ye et al., 2023; Alayrac et al., 2022; Zhu et al., 2023; Dai et al., 2023; Li et al., 2022), thereby accelerating the development of Multimodal Large Language Models (MLLMs) and further bringing us closer to the realization of Artificial General Intelligence (AGI).

Current MLLMs mainly rely on pre-trained LLMs and vision encoders, focusing on bridging the modality gap by aligning visual features with language embedding tokens. Existing research predominantly employs either a Q-Former (Li et al., 2023b; Bai et al., 2023b) or an MLP projector (Liu et al., 2023b; Chen et al., 2023b) to align vision encoders with LLMs. The training data mainly consists of English-centric data from image captions and multimodal conversations. During training, the alignment component converts the visual features into language embedding tokens. By incorporating encoded visual features, the LLM can integrate visual information to respond to multimodal inputs, thereby enabling the model to possess visual question answering and reasoning capabilities.

Multilingual capability in MLLMs entails the ability to generate responses in the same language as the input, catering to the linguistic diversity inherent in conversation. Therefore, it is critically important for processing language-specific content and cultural differences, ensuring equitable access to technological benefits for individuals across diverse regions and nations (Chen et al., 2022; Hu et al., 2023). Many LLMs possess multilingual capabilities (Touvron et al., 2023; Bai et al., 2023a;

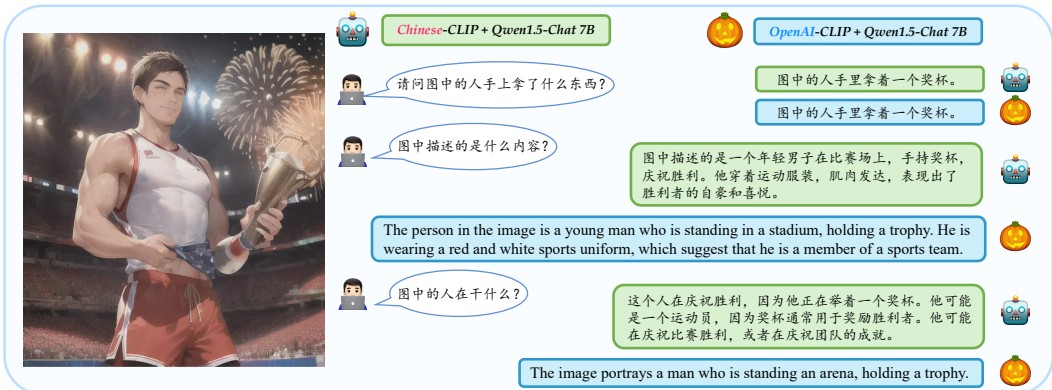

Figure 1: The output of OpenAI-CLIP-based and Chinese-CLIP-based models using the same Chinese prompts. We can observe that the OpenAI-CLIP-based model exhibits confusion between Chinese and English responses.

OpenAI, 2023b), enabling diverse language responses according to user input. However, after the alignment training of MLLMs, the model may lose its ability to understand, process, or generate in non-English languages, and we call this phenomenon *multilingual erosion*. For example, LLaVA (Liu et al., 2023b) usually responds in English, regardless of the input language. Therefore, it is essential to enhance MLLM's multilingual capabilities during multimodal alignment.

The main reason for multilingual erosion is that the data used for multimodal alignment is highly imbalanced at the language level. Due to the dominance of English-centric data, while the model aligns visual and textual tokens well in English, it performs poorly in other languages. Hence, it is crucial to align visual and textual tokens compatibly at the language level. We hypothesize that *multilingual erosion* may arise from the lack of alignment between visual tokens and textual tokens in other languages. From the perspective of pre-trained datasets, OpenAI-CLIP (Radford et al., 2021) is trained on the large-scale image-text pairs through contrastive learning, with the text corpus being mostly in English, potentially biasing image encoding towards an English semantic space. As shown in Figure 1, we train two separate models using the same data: one with OpenAI-CLIP vision encoder and the other with Chinese-CLIP (Yang et al., 2022) vision encoder. Interestingly, the model equipped with OpenAI-CLIP struggles to generate suitable outputs according to Chinese inputs, while the other model with Chinese-CLIP can not only understand the queries but also generate appropriate outputs in Chinese. Furthermore, we observed a performance improvement, from 66.4 to 68.3, on the MMBench-CN (Liu et al., 2023c) dataset when using Chinese-CLIP. Therefore, the challenge arises: how to use English-centric multilingual image-text data to bridge the modality gap while enhancing the MLLM's multilingual capabilities.

Due to the scarcity of non-English multimodal data (*e.g.*, lack of large-scale, high-quality image-text data), we require almost the same amount of image-text data as LLaVA to enhance the model's multilingual capabilities. Moreover, motivated by preliminary experiments, it is necessary to condition the visual tokens on diverse language inputs. In this paper, we introduce PARROT, a novel method that utilizes textual guidance to drive visual token alignment at the language level and converts visual tokens into language-specific embeddings using a Mixture-of-Experts (MoE) module (Jacobs et al., 1991; Shazeer et al., 2017). Specifically, we first calculate the cross-attention between the class token of visual features extracted by the vision encoder and the text embeddings derived from word token embeddings. The result is then passed through the router of MoE to obtain the activated probability distribution of each language expert. Subsequently, demanding the input language, the English-biased visual tokens are converted into language-specific embeddings using the selected experts. This enables PARROT not only to enhance its multilingual capabilities but also to bridge the multimodal gap effectively.

To address the scarcity of current multilingual benchmarks, we introduce a new benchmark encompassing six languages: English, Chinese, Portuguese, Arabic, Turkish, and Russian. This includes an extension of the MMBench-DEV dataset to these six languages and a Massive Multilingual Multimodal Benchmark (**MMMB**) featuring 2,000 evaluation questions per language, totaling 12,000 questions. Through a semi-automatic approach, which is shown in Figure 3, we alleviate the potential introduction of noise and errors when constructing the benchmark. To comprehensively assess our model's capabilities, we compare several open-source multimodal methods and evaluate some

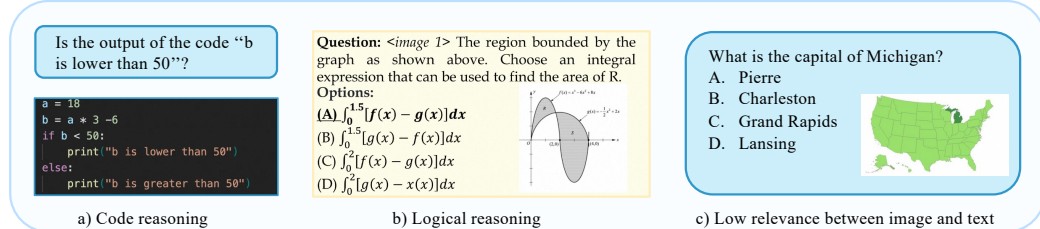

Figure 2: Some bad cases for the existing multilingual benchmark. **Left:** code reasoning is strongly related to English. **Middle:** logical reasoning is too challenging. **Right:** lack relevance between image and text.

proprietary models. Extensive experiments validate the PARROT's state-of-the-art performance across two multilingual benchmarks. Specifically in Turkish and Arabic, our method even outperforms LLaVA-NeXT (Liu et al., 2024) by more than 10 percentage points in both benchmarks. Additionally, we evaluate our model across a broad range of multimodal benchmarks (*e.g.*, MME (Fu et al., 2023), ScienceQA-IMG (Lu et al., 2022), and SEED-Bench-IMG (Li et al., 2024a)), demonstrating its competitive performance in diverse tasks.

**Related Work.** 1) Multimodal Large Language Models. Current MLLMs typically consist of a vision encoder, LLM, and fusion module. LLaVA (Liu et al., 2023b) uses a simple MLP projector to connect the vision encoder and LLM. BLIP2 (Li et al., 2023b) and InstructBLIP (Dai et al., 2023) employ Q-Former to bridge the modality gap. GPT-4o (OpenAI, 2024), Gemini (Reid et al., 2024), and Claude3 (anthropic, 2024) has achieved impressive results. 2) Multilingual Multimodal Models. mCLIP (Chen et al., 2023a), PaLI (Chen et al., 2022), and VisCPM (Hu et al., 2023) endow models with multilingual capabilities. A detailed related work is presented in Appendix B.

## 2 MMMB: A MASSIVE MULTILINGUAL MULTIMODAL BENCHMARK

In this section, we first discuss the limitations of existing benchmarks and then present the characteristics that an ideal multilingual benchmark should possess. Furthermore, we design and construct a new benchmark and provide its corresponding evaluation strategy.

### 2.1 LIMITATIONS OF EXISTING BENCHMARKS

There are several existing multilingual benchmarks (*e.g.*, Multi30K (Elliott et al., 2016), M3Exam (Zhang et al., 2024b), MMBench (Liu et al., 2023c), and LLaVA-Bench (Liu et al., 2023b; Hu et al., 2023)) for MLLMs, but they have some limitations: **1) Outdated Benchmarks.** Multi30k is designed for image-text retrieval tasks, and the performance has nearly reached the upper bound due to the relatively easy problems. **2) Non-Standardized Evaluations.** Other benchmarks, like LLaVA-Bench, rely on evaluations using GPT-4. Dependence on GPT-4 as a de facto "Ground Truth" may hinder reproducibility. Meanwhile, since LLaVA uses a deprecated version (GPT-4-0314), using other different versions could result in unfair comparisons. On the other hand, because M3Exam does not offer consistent test samples across different languages, it cannot ensure whether poor performance is due to the problem's difficulty or the model's lack of multilingual capabilities. **3) Limited Languages.** MMBench and LLaVA-Bench are limited in English and Chinese, which can not measure the multilingual capabilities across a broad spectrum.

### 2.2 CHARACTERISTICS OF AN EFFECTIVE MULTILINGUAL BENCHMARK

To more suitably evaluate the multilingual capabilities of MLLMs, an ideal benchmark should exhibit the following characteristics:

**1) Languages with Significant Differences.** It should cover a diverse array of language families, selecting languages that are as distinct and non-repetitive as possible. This ensures a broad assessment of MLLMs' ability to adapt across linguistic variances.

**2) Problems with Medium Level of Difficulty.** The problems should not be too difficult (*e.g.*, logical reasoning) because the aim is to assess the multilingual understanding, processing, and generating capabilities of MLLMs, not logical reasoning skills.

**3) Tasks with Multilingual and Multimodal.** As shown in Figure 2, data within datasets should not be strongly related to English (*e.g.*, code reasoning). It cannot be inherently transformed into multiple languages since they are composed of English words. Moreover, images should be an indispensable part when MLLMs answer the question. For instance, if given a map of the United States and asked to identify its capital, MLLMs only require the text-only ability to answer this question. Therefore, it is essential that questions highlight a significant correlation between images and texts.

**4) Content Consistency across Languages.** The goal of this benchmark is to evaluate the multilingual capabilities of MLLMs, and we aim to show the discrepancies across different languages fairly. For example, if English questions mainly focus on *addition within one hundred* while Chinese questions mainly concern *calculus computation*, it becomes difficult to ascertain whether poor performance in Chinese arises from the complexity of the problem or the limited multilingual capabilities of MLLMs. Hence, it is crucial to ensure content consistency across languages for a fair comparison.

## 2.3 CONSTRUCTION OF THE MULTILINGUAL BENCHMARK

We select six languages for inclusion: English (*en*), Chinese (*zh*), Portuguese (*pt*), Arabic (*ar*), Turkish (*tr*), and Russian (*ru*). These languages represent a diverse range of linguistic families, and we list the detailed information and some multilingual cases in Figure 4. In terms of dataset requirements and consistency, our benchmark incorporates datasets in two main respects: **1)** Since MM-Bench (Liu et al., 2023c) officially includes English and Chinese versions, we extend it to the other four languages. **2)** For the massive multilingual multimodal benchmark, denoted as **MMMB**, we select and clean the suitable data from ScienceQA (Lu et al., 2022),

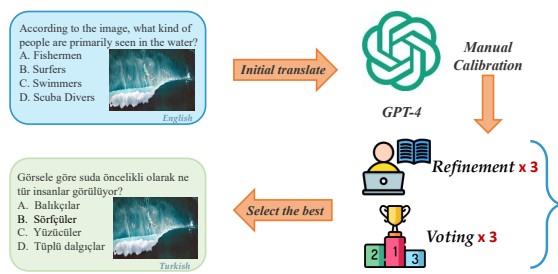

Figure 3: The calibration process for constructing a multilingual benchmark. The calibration process is mainly divided into two stages: GPT-4 Translation and Manual Calibration.

MME (Fu et al., 2023), and SEED-Bench (Li et al., 2024a) datasets with established guidelines. These datasets are then processed into a Visual Question Answering format, resulting in a total of 12,000 samples across all six languages.

To alleviate the potential introduction of noise and errors through the data acquisition process, we employ the following strategies to enhance the quality of our translations in Figure 3. First of all, we choose GPT-4 to translate the original problem into the target language. Then, we input the first translation result back into GPT-4 for a re-check and refinement. This step helps to identify and correct any immediate errors or inconsistencies in the translation. For manual calibration, we engage two groups of professional translators for each language involved in the study:

**1) First Group for Refinement.** This group consists of three language experts who independently review and refine the translations produced by GPT-4. This process results in three distinct translation versions for each piece of content.

**2) Second Group for Voting.** The second group of experts is responsible for evaluating these three refined translations. Through a voting process, they will choose the best translation that accurately captures the intended meaning and nuances of the original text.

This calibration process significantly enhances the data quality by reducing errors and ensuring that translations are contextually appropriate across different languages. As a result, our benchmark reflects a better level of linguistic precision and cultural relevance, which we believe contributes positively to the overall robustness of our research findings. In future versions, we will include more detailed information to enhance readability and completeness.

## 2.4 EVALUATION STRATEGY

Since random guessing can lead to ∼25% Top-1 accuracy for 4-choice questions, potentially reducing the discernible performance differences between various MLLMs. Additionally, MLLMs may prefer to predict a certain choice among all given choices (Liu et al., 2023c), which further amplifies the bias

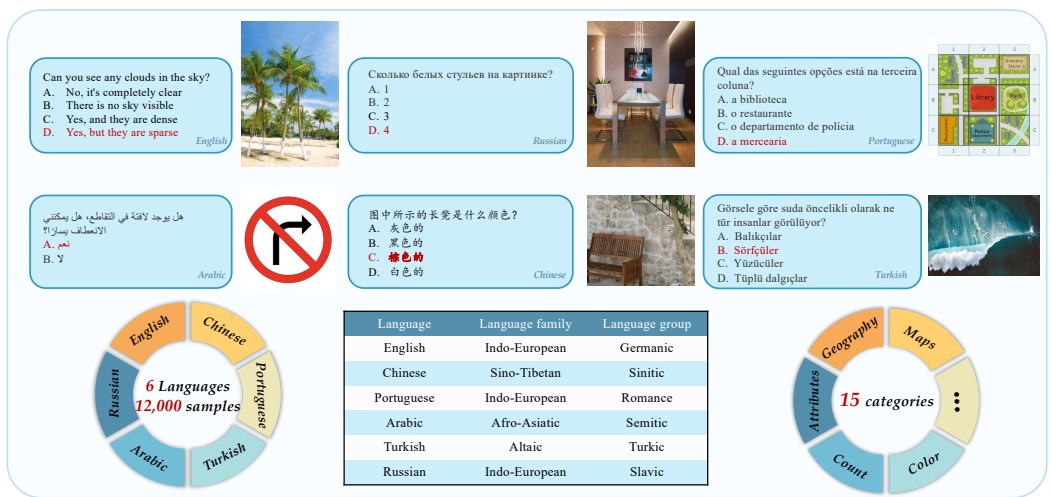

Figure 4: Overview of MMMB. It incorporates 6 languages, 15 categories, and 12,000 questions.

in evaluation. To address these issues, we implement a circular validation strategy inspired by MM-Bench. Specifically, MMMB is adapted to the format of Yes/No questions, where each image is paired with two questions, demanding 'Yes' and 'No' answers, respectively. As shown in Figure 9, an answer is considered accurate only if both questions are answered correctly; failing either result in marking the entire instance as incorrect. This strategy ensures a more rigorous evaluation of MLLMs, reducing the likelihood of random guessing and promoting more validated comparisons across different models.

## 3 METHODS

### 3.1 PRELIMINARIES: VISUAL INSTRUCTION TUNING

A representative work in MLLMs is LLaVA (Liu et al., 2023b), which introduces a simple yet effective method for achieving alignment between the vision encoder and the pre-trained LLM. Specifically, for a given input image $\mathbf{X}_\text{v}$, LLaVA utilizes the pre-trained CLIP vision encoder ViT-L/14 (Radford et al., 2021) to extract the visual features $\mathbf{Z}_\text{v} = g(\mathbf{X}_\text{v})$. It then employs Vicuna (Chiang et al., 2023) as the LLM to obtain the textual embeddings $\mathbf{H}_t$. To align the vision encoder with the LLM, a projector in the form of a multi-layer perceptron (MLP) denoted as $\mathbf{W}$ is learned. This projector converts $\mathbf{Z}_\text{v}$ into language embedding tokens $\mathbf{H}_\text{v}$, effectively facilitating the integration of multimodal information within the LLM's framework.

$$\mathbf{H}_\text{v} = \mathbf{W} \cdot \mathbf{Z}_\text{v}, \text{ with } \mathbf{Z}_\text{v} = g(\mathbf{X}_\text{v}). \tag{1}$$

Finally, we input $\mathbf{H}_v$ and $\mathbf{H}_t$ into LLM to generate the model's responses. However, after the modality alignment training, LLaVA loses its ability to process in non-English languages.

### 3.2 PILOT STUDY

To address the challenge of multilingual erosion in MLLMs due to the dominance of English in image-text data, we hypothesize that there is an inherent mismatch between visual tokens $\mathbf{H}_v$ and textual tokens $\mathbf{H}_t$, which tends to bias them towards English semantics, making the model more likely to generate outputs in English. Specifically, the widely-used vision encoder of OpenAI-CLIP (Radford et al., 2021) is pre-trained on a vast corpus of English-centric image-text pairs, resulting in visual representations more aligned with English.

To explore this phenomenon, we train two distinct models using the same data: one incorporating OpenAI-CLIP vision encoder and the other integrating Chinese-CLIP (Yang et al., 2022) vision encoder. As shown in Figure 1, the model equipped with OpenAI-CLIP struggles to generate suitable outputs according to the Chinese inputs, whereas the model using Chinese-CLIP not only understands

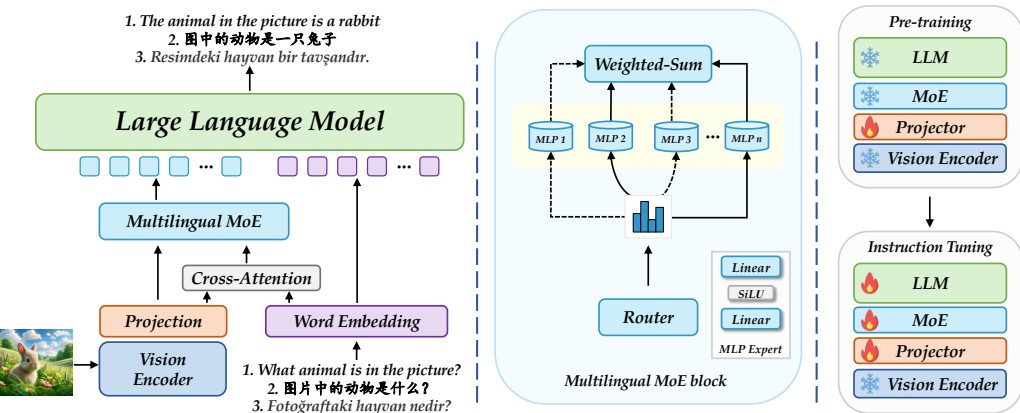

Figure 5: The overall architecture of PARROT. It converts English-biased features to language-specific features based on the multilingual MoE module, aiming to improve the multilingual capabilities. The training details within each stage are presented on the right.

the queries but also generates appropriate outputs in Chinese. Moreover, to further evaluate the model's Chinese capability, we test it on Chinese datasets and observe improved performance: from 66.4 to 68.3 on MMBench-CN and from 62.4 to 66.1 on MMMB-zh when utilizing Chinese-CLIP.

### 3.3 TEXTUAL GUIDANCE TO DRIVE VISUAL TOKEN ALIGNMENT

Due to the low-resource nature of non-English multimodal data (*e.g.*, lack of large-scale, high-quality image-text data), we need to use nearly the same amount of data as LLaVA to enhance the model's multilingual capabilities. Furthermore, motivated by these interesting findings and aiming to align visual tokens with textual embeddings at the language level, we propose PARROT, a novel approach that leverages textual guidance to facilitate the multilingual alignment of visual features. PARROT enables the transition of English-biased visual features acquired through the OpenAI-CLIP to accommodate other languages. This approach ensures that it can provide language-specific visual tokens to LLM based on the multiple language inputs, thereby enhancing its multilingual capabilities.

First, we extract visual features through the vision encoder and transform them into language embedding tokens $\mathbf{H}_v$ using a projector. We obtain the embeddings $\mathbf{H}_t \in \mathbb{R}^{N \times C}$ derived from text inputs via the word embedding table. Subsequently, to convert the English-biased features into language-specific features using textual guidance, we employ a cross-modal cross-attention mechanism to obtain $\mathbf{H}_v' \in \mathbb{R}^C$:

$$\mathbf{H}_v' = \text{Attention}(\mathbf{Q}, \mathbf{K}, \mathbf{V}) = \text{Softmax}\left(\frac{\mathbf{H}_v^{\text{cls}}\mathbf{H}_t^T}{\sqrt{C}}\right)\mathbf{H}_t, \tag{2}$$

where $\mathbf{Q}$ equals the matrix $\mathbf{H}_v$, $\mathbf{K}$ and $\mathbf{V}$ are equivalent to $\mathbf{H}_t$. $\mathbf{H}_v^{\text{cls}} \in \mathbb{R}^C$ is the [CLS] token of $\mathbf{H}_v$. Based on the multilingual inputs, this process allows the visual features to be dynamically adjusted and transformed to the language-specific semantic embeddings.

Since the projected language embedding tokens $\mathbf{H}_v$ are English-biased, we need to convert them to language-specific embeddings for different languages. To this end, we introduce a lightweight Mix-of-Experts (MoE) module, which includes a router and several language transformation experts. The router of MoE is a linear layer that generates a probability distribution over the set of experts $\mathcal{E} = [e_1, e_2, \cdots, e_E]$, effectively predicting the probability of selecting and activating each expert. Each expert is an MLP designed to convert English-biased embeddings into language-specific embeddings. The inputs to experts $\mathcal{E}$ is $\mathbf{H}_v$, and the outputs have the same dimensions as the inputs.

Subsequently, to obtain a normalized probability distribution for activating language-specific experts, $\mathbf{H}_v'$ is fed as input to the router. The router network contains a linear layer that computes the normalized weight matrix using $\mathbf{H}_v'$ for voting, producing $\mathcal{P} \in \mathbb{R}^E$:

$$\mathcal{P} = \text{Softmax}(\text{Linear}(\mathbf{H}_v')), \tag{3}$$

which selects and activates the specific experts. Moreover, we process the English-biased embeddings $\mathbf{H}_v$ through the selected experts to convert them into language-specific visual representations:

$$\text{MoE}(\mathbf{H}_v) = \sum_{i=1}^{k} \mathcal{P}[i] \cdot \mathcal{E}(\mathbf{H}_v)_i. \tag{4}$$

This approach effectively aligns English-biased embeddings with multiple languages, ensuring a more accurate and comprehensive representation across different linguistic contexts. To stabilize training and convert visual embeddings with less variance in visual-semantic information, ensuring the model performs well in tasks beyond the multilingual multimodal domain, we utilize MoE reweighting to obtain the final language-specific visual embeddings $\mathbf{G}_v$:

$$\mathbf{G}_v = \mathbf{H}_v + \alpha\text{MoE}(\mathbf{x}), \tag{5}$$

where $\alpha$ is the trade-off parameter. In conclusion, we first fuse the visual and textual inputs via Eq. 2 to transform the visual embeddings with textual guidance. Moreover, the fused result is inputted into the MoE module to select and activate the most relevant language experts via Eq. 3 and then obtain the language-specific embeddings via Eq. 4. Lastly, we employ MoE reweighting to convert visual embeddings with less variance in original visual-semantic information 5. This approach enables us to endow the MLLM with multilingual capabilities using as little multilingual data as possible. Figure 5 illustrates the architecture, the detailed MoE module, and the training stages of PARROT.

## 3.4 TRAINING STAGE

Our goal is to utilize as little multilingual data as possible to enhance the multilingual capabilities of MLLMs. The whole training procedure is divided into two distinct stages:

**Stage 1: Modality Alignment.** In this stage, we keep both the vision encoder and the LLM weights frozen, focusing solely on optimizing the projectors to align the visual features $\mathbf{H}_v$ with the pre-trained LLM word embedding. This stage can be likened to training a visual tokenizer that is compatible with the frozen LLM. To enhance the diversity of images, we extract a portion of data from LAION (Schuhmann et al., 2022) and CC12M (Changpinyo et al., 2021) datasets and construct the in-house caption data through GPT-4V.

**Stage 2: Instruction Tuning for Multilingual Alignment.** We still keep the vision encoder weights frozen while continuing to train the projector, MoE, and LLM. Due to the design of the MoE module, PARROT can rapidly learn to align visual representations across multiple languages by using a small amount of multilingual image-text data. As shown in Table 5, we only use nearly 10K training data for each language in stage 2. This approach is particularly beneficial given the scarcity of data resources in low-resource languages.

To address the challenge of limited data in non-English languages, we use a semi-automatic approach similar to the one depicted in Figure 3 to acquire image-text data. Initially, we partition the ShareGPT4V dataset (Chen et al., 2023b) randomly for each language, extracting a selection of non-duplicate, non-parallel image-text data for training. Subsequently, we implement a translation and calibration scheme using GPT-4 to convert English texts into texts of other languages. Recognizing that this step may introduce noise and potential translation errors, we apply a manual calibration process to further fine-tune and clean the data, thereby obtaining high-quality multilingual image-text data. This two-stage training approach ensures effective modality and multilingual alignment, even with limited non-English data, aligning well with the realities of data scarcity in low-resource languages.

## 4 EXPERIMENTS

In this section, we begin with an overview of the experimental framework, providing details on specific implementations, evaluation benchmarks, and MLLMs used for comparative evaluation. Following this, we conduct a comprehensive comparison of PARROT with the state-of-the-art approaches using multilingual benchmarks. Additionally, we compare PARROT with leading models across a range of multimodal tasks. Finally, this section concludes with ablation studies and visualization of multilingual cases, highlighting the exceptional ability of PARROT in handling multilingual tasks.

## 4.1 EXPERIMENTAL SETUP

**Implementation Details:** In this study, we configure PARROT with the pre-trained CLIP ViT-L/14 (Radford et al., 2021) as the vision encoder and Qwen1.5-Chat (Bai et al., 2023a) as the backbone for LLM. The initial learning rates for the two stages are set at $1e^{-3}$ and $2e^{-5}$, respectively, with the batch size of 256 and 128. The entire training process is notably optimized to 21 hours on the $16\times$A100 GPUs setup, attributed to the use of the relatively small training datasets. Additionally, BF16 and TF32 precision formats are employed to meticulously balance speed and accuracy throughout the training process. As defined in Eq. 4, we set the number of experts to six to match the number of languages. Each expert is an MLP composed of two linear layers with SiLU (Elfwing et al., 2018) activation function. More details are shown in Table 4.

**Evaluation Benchmark:** Our evaluation is divided into two parts: one evaluates the multilingual capabilities of MLLMs, while the other assesses its overall performance. The first evaluation is performed on two datasets: MMBench (Liu et al., 2023c) and a newly developed benchmark MMMB. For MMBench, we expand it to include six languages through translation via GPT-4, followed by manual verification. For MMMB, we construct it following the methodology described in Section 2. We present the accuracy for each language in Table 1. Furthermore, the second evaluation covers a wide broad range of multimodal tasks, such as MME (Fu et al., 2023), MMStar (Chen et al., 2024b), ScienceQA (Lu et al., 2022), RealWorldQA (x.ai, 2024) and SEED-Bench (Li et al., 2024a), with performance reported using a radar chart in Figure 6b.

**Comparison Models:** For comprehensive comparisons, we select leading open-source models in MLLMs, including LLaVA-1.5 (Li et al., 2023a), LLaVA-NeXT (Liu et al., 2024), Qwen-VL (Bai et al., 2023b), Monkey (Li et al., 2023d), VisualGLM (Du et al., 2022), VisCPM (Hu et al., 2023), MiniGPT-4-v2 (Zhu et al., 2023), ShareGPT4V (Chen et al., 2023b), InstructBLIP (Dai et al., 2023), mPLUG-Owl2 (Ye et al., 2023), Mini-Gemini (Li et al., 2024c). Furthermore, we incorporate closed-source methods in our benchmarks, including GPT-4V (Chen et al., 2023b), Qwen-VL-MAX (Bai et al., 2023b), and Gemini Pro (Reid et al., 2024), to demonstrate their remarkable performance. For the evaluation process, we employ the VLMEvalKit in OpenCompass (Duan et al., 2024), ensuring consistent configuration settings across all methods to maintain fairness in comparison. For most of the mentioned methods, we directly use the VLMEvalKit implementation. Alternatively, we integrate other methods not officially provided into this framework to ensure consistency in evaluation.

## 4.2 MAIN RESULTS

In this section, we present the results of the multilingual experiment in Table 1 and the overall experiment in Figure 6b. According to Table 1, PARROT-14B achieves state-of-the-art (SOTA) performance in all languages on the MMBench benchmark and also achieves the SOTA performance in five languages on the MMMB benchmark, with English in the second place. The multilingual capabilities of PARROT-14B closely reach that of GPT-4V, demonstrating the exceptional ability of our proposed architecture. Notably, PARROT-7B also validates the SOTA performance on both benchmarks across all languages, even surpassing the LLaVA-NeXT-13B. Additionally, as shown in Figure 6b, this evaluation aims to show that PARROT not only possesses excellent multilingual capabilities but also provides an overall understanding of PARROT's capabilities in handling various complex multimodal tasks (*e.g.*, MME (Fu et al., 2023), MMStar (Chen et al., 2024b), and SEED-Bench (Li et al., 2024a)). Additionally, as depicted in Figure 6c, we visualize the expert distributions within the MoE. It is evident that the second expert is predominantly activated when using the Chinese prompt, indicating that different experts are utilized for various language prompts. In existing multilingual benchmarks, PARROT also demonstrates competitive performance while using less than 1% of the data compared to other multilingual MLLMs, as illustrated in Figure 6.

## 4.3 ABLATION STUDY

In this section, we present an ablation study to examine the contribution of individual components to the overall performance of PARROT. Additionally, we will demonstrate the impact of incorporating training datasets in various languages on multilingual performance.

**Ablation study on each component.** We conduct an ablation experiment on the multilingual data and the MoE module. As shown in Figure 6a, using multilingual data improves performance in

Table 1: Accuracy performance comparison on multilingual benchmarks. We report all compared methods with VLMEvalKit (Duan et al., 2024). The best and second results are shown in **bold** and underline, respectively.

| Method | LLM | MMMB | | | | | | MMBench | | | | | |
|---|---|---|---|---|---|---|---|---|---|---|---|---|---|
| | | *en* | *zh* | *pt* | *ar* | *tr* | *ru* | *en* | *zh* | *pt* | *ar* | *tr* | *ru* |
| *Open-source models* | | | | | | | | | | | | | |
| LLaVA-1.5 (Liu et al., 2023a) | Vicuna-v1.5-7B | 67.07 | 58.83 | 59.76 | 43.50 | 46.43 | 59.06 | 65.37 | 58.33 | 59.02 | 36.16 | 43.90 | 56.95 |
| LLaVA-1.5 (Liu et al., 2023a) | Vicuna-v1.5-13B | 69.76 | 62.86 | 60.76 | 45.49 | 54.44 | 62.69 | 68.98 | 63.23 | 62.97 | 46.56 | 53.17 | 61.59 |
| LLaVA-NeXT (Liu et al., 2024) | Vicuna-v1.5-7B | 70.87 | 61.57 | 61.81 | 42.74 | 46.95 | 63.85 | 67.95 | 60.56 | 60.39 | 38.40 | 45.36 | 59.62 |
| LLaVA-NeXT (Liu et al., 2024) | Vicuna-v1.5-13B | **74.44** | 67.19 | 63.21 | 45.36 | 53.09 | 68.24 | 70.87 | 64.51 | 64.08 | 45.36 | 52.92 | 61.85 |
| Qwen-VL (Bai et al., 2023b) | Qwen-7B | 52.63 | 36.37 | 38.65 | 36.54 | 37.42 | 40.70 | 42.26 | 22.25 | 25.08 | 18.72 | 26.37 | 28.17 |
| Qwen-VL-Chat (Bai et al., 2023b) | Qwen-7B | 56.02 | 57.77 | 46.37 | 43.04 | 41.05 | 48.65 | 54.29 | 56.52 | 43.12 | 35.73 | 39.17 | 42.86 |
| MiniGPT-4-v2 (Zhu et al., 2023) | LLaMA2-13B | 38.71 | 30.05 | 31.52 | 26.60 | 26.02 | 29.23 | 23.88 | 11.76 | 14.26 | 2.49 | 6.78 | 12.54 |
| ShareGPT4V (Chen et al., 2023b) | Vicuna-v1.5-7B | 69.24 | 60.23 | 60.29 | 43.57 | 45.26 | 61.23 | 69.59 | 61.6 | 59.62 | 37.37 | 43.38 | 59.45 |
| InstructBLIP (Dai et al., 2023) | Vicuna-7B | 39.47 | 32.92 | 35.67 | 23.80 | 28.36 | 36.37 | 27.83 | 18.81 | 27.14 | 3.26 | 8.50 | 20.87 |
| mPLUG-Owl2 (Ye et al., 2023) | LLaMA2-7B | 67.25 | 60.99 | 59.70 | 45.78 | 45.43 | 62.63 | 66.15 | 59.36 | 58.24 | 37.88 | 47.68 | 60.39 |
| Monkey (Li et al., 2023d) | Qwen-VL-7B | 66.02 | 58.18 | 46.31 | 38.83 | 37.66 | 48.59 | 58.07 | 53.52 | 49.57 | 31.01 | 31.35 | 45.18 |
| Monkey-chat (Li et al., 2023d) | Qwen-VL-7B | 71.63 | 66.54 | 60.35 | 48.77 | 46.31 | 58.59 | 70.79 | 65.72 | 65.03 | 46.90 | 48.10 | 59.36 |
| VisualGLM (Du et al., 2022) | ChatGLM-6B | 31.05 | 18.07 | 19.42 | 15.38 | 22.81 | 19.77 | 23.2 | 17.18 | 11.43 | 2.92 | 6.62 | 5.33 |
| VisCPM-Chat (Hu et al., 2023) | CPM-Bee-10B | 53.10 | 47.54 | 28.19 | 26.90 | 26.78 | 26.84 | 45.88 | 46.39 | 15.81 | 1.46 | 9.19 | 1.20 |
| PARROT | Qwen1.5-7B | 70.00 | 68.13 | 67.31 | 62.69 | 58.01 | 66.26 | 70.70 | 70.36 | 65.12 | 57.82 | 58.43 | 64.00 |
| PARROT | Qwen1.5-14B | 73.92 | 71.64 | 69.82 | 68.13 | 64.33 | 70.18 | 74.40 | 72.25 | 69.16 | 66.15 | 64.52 | 69.33 |
| *Closed-source models* | | | | | | | | | | | | | |
| GPT-4V (OpenAI, 2023b) | Private | 74.97 | 74.21 | 71.46 | **73.51** | 68.95 | 73.10 | **77.60** | 74.40 | 72.51 | 72.34 | **70.53** | 74.83 |
| Gemini Pro (Team et al., 2023) | Private | 75.03 | 71.87 | 70.64 | 69.94 | **69.59** | 72.69 | 73.63 | 72.08 | 70.27 | 61.08 | 69.76 | 70.45 |
| Qwen-VL-MAX (Bai et al., 2023b) | Private | **77.19** | 75.26 | 72.16 | 70.82 | 66.02 | **74.21** | 76.80 | **77.58** | 74.57 | 75.00 | 69.07 | 75.00 |

(a) Ablation study.  (b) Multiple multimodal tasks.  (c) Expert distributions.

Figure 6: **Left:** The ablation study of multilingual data and the MoE module using the MMBench benchmark. **Middle:** The performance of PARROT on a broad range of multimodal tasks compared with existing models. Models with 7B parameters are used for the two experiments. **Right:** Expert distributions of MoE. We summarize the activated experts during the feed-forward process using Chinese Prompts.

each language. Moreover, the MoE module significantly improves performance, demonstrating the effectiveness of our proposed method.

**Ablation study on different datasets.** As shown in Table 2, it is evident that the inclusion of different multilingual datasets continually improves performance on the MMBench benchmark, and all models with 7B parameters are used for this experiment. This highlights the robustness and scalability of our approach to handling multiple languages effectively. We also conduct an ablation study using different multilingual datasets in Table 10.

**Ablation study on monolingual fine-tuning datasets.** The ablation study presented in Table 16 evaluates the performance of different monolingual datasets added incrementally to the baseline dataset LLaVA-1.5-finetune. It highlights the significant impact of adding different multilingual datasets to a baseline model. Each dataset incrementally improves performance in its respective language and, when combined, leads to overall enhanced performance across all evaluated languages.

Table 2: Ablation study on different multilingual training datasets in MMBench benchmark. Models with 7B parameters are used for this ablation.

| Dataset | **English** | | **Chinese** | | **Portuguese** | | **Arabic** | | **Turkish** | | **Russian** | |
|---|---|---|---|---|---|---|---|---|---|---|---|---|
| LLaVA-1.5-finetune | 69.4 | | 66.6 | | 60.3 | | 55.3 | | 52.1 | | 60.7 | |
| + *zh* | 69.2 | -0.2 | 68.6 | +2.0 | 64.1 | +3.8 | 59.1 | +3.8 | 50.9 | -1.2 | 61.6 | +0.9 |
| + *zh pt* | 71.1 | +1.7 | 70.4 | +3.8 | 65.4 | +5.1 | 57.9 | +2.6 | 52.1 | +0.0 | 62.9 | +2.2 |
| + *zh pt ar* | 71.0 | +1.6 | 68.6 | +2.0 | 65.7 | +5.4 | 58.6 | +3.3 | 52.2 | +0.1 | 62.2 | +1.5 |
| + *zh pt ar tr* | 70.4 | +1.0 | 68.7 | +2.1 | 64.9 | +4.6 | 61.2 | +5.9 | 59.7 | +7.6 | 62.0 | +1.3 |
| + *zh pt ar tr ru* | 70.7 | +1.3 | 70.4 | +3.8 | 65.1 | +4.8 | 57.8 | +2.5 | 58.4 | +6.3 | 64.0 | +3.3 |

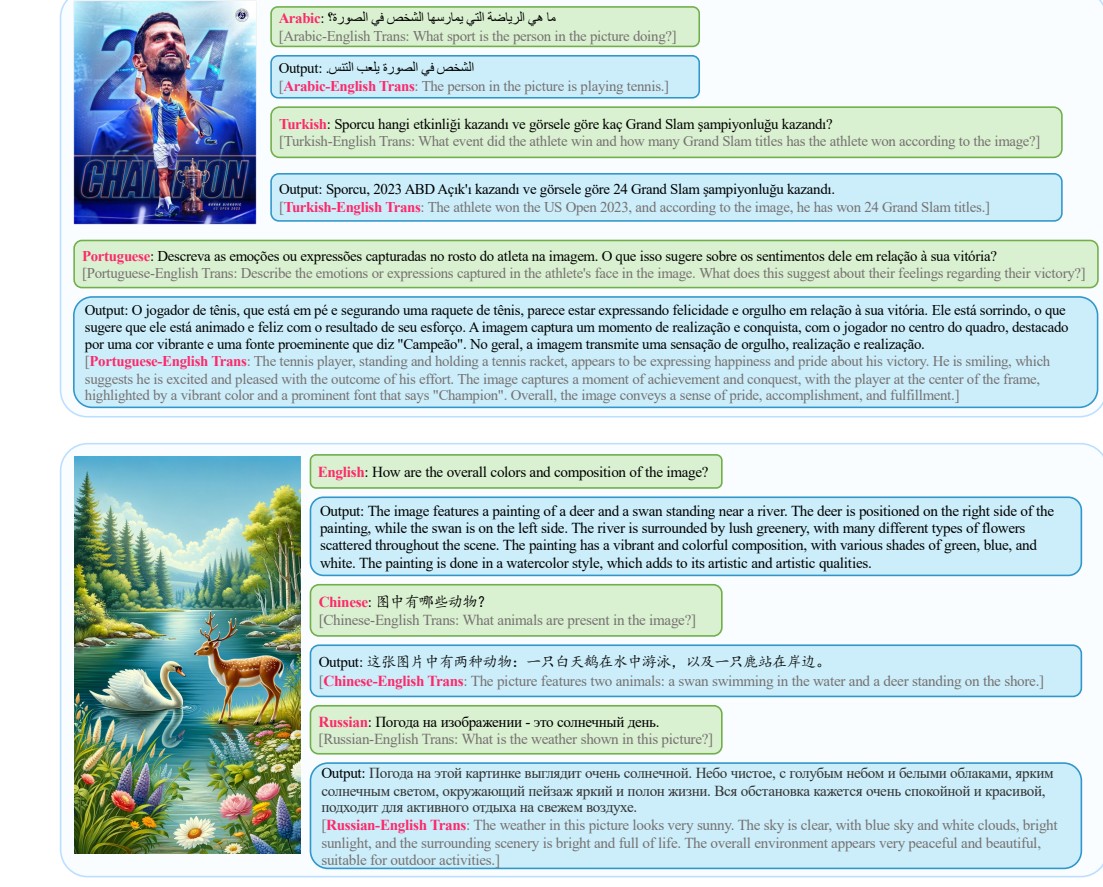

Figure 7: Multimodal conversation cases of PARROT in multiple languages.

This indicates the robustness and effectiveness of the proposed method in handling multilingual data, making it a scalable solution for multilingual tasks.

## 4.4 VISUALIZATION OF MULTILINGUAL CONVERSATIONS

To enhance the intuitive understanding of the PARROT's multilingual capability, we prepare a comprehensive case study accompanied by illustrative visuals. For instance, as depicted in Figure 7, our framework demonstrates remarkable multilingual capabilities. This underscores the PARROT's versatility in navigating different languages and presents its potential in bridging linguistic gaps across diverse domains. Through careful analysis and visualization, we aim to provide a deeper insight into the mechanism driving this capability, illustrating its practical implications and potential applications in real-world scenarios. This visualization serves as a strong indicator of the PARROT's solid architecture and its exceptional ability to understand, process, and generate multiple languages with remarkable efficiency. More multilingual conversation cases are shown in Appendix H.

## 5 CONCLUSION

This paper addresses the critical challenge of enhancing the multilingual capabilities of MLLMs. We introduce PARROT, a novel method leveraging textual guidance to drive visual token alignment at the language level, thus enabling the transition of English-biased visual embeddings into language-specific ones using an MoE module. Extensive experiments conducted on a newly introduced Massive Multilingual Multimodal Benchmark (MMMB) across six languages demonstrate the state-of-the-art performance of PARROT compared to existing methods, particularly presenting remarkable improvements in Turkish and Arabic. Furthermore, our model exhibits competitive results across a wide range of diverse multimodal benchmarks, emphasizing its efficacy in addressing both multilingual and multimodal challenges. PARROT not only advances the frontier of MLLMs but also underscores the importance of equitable access to technological benefits across linguistic and cultural diversities.

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

# A    MORE DETAILS OF TRAINING DATASETS

In this section, we analyze the multilingual data in LLaVA (Liu et al., 2023b). From Table 3 and Figure 8, it is evident that during the pre-train stage, LLaVA solely utilizes multimodal image-text pairs data for training, comprising 558K of English data. During the SFT stage, both multimodal and text-only data are incorporated into the training process. Multilingual data appear only in the text-only dataset. Apart from English, the most prominent non-English data is Chinese, amounting to just 3.1K, constituting 0.25% of the total dataset. Therefore, it is evident that LLaVA's datasets are English-centric and imbalanced. The specific language and abbreviation are as follows: English (*en*), Chinese (*zh*), Korean (*ko*), Spanish (*es*), French (*fr*), Japanese (*ja*), German (*de*), Portuguese (*pt*), Traditional Chinese (*zh-tw*), Italian (*it*).

Table 3: The detailed information about LLaVA's datasets.

(a) The language information in two stages.

| Training Stage | Type | Total Size | English | Other Languages |
|---|---|---|---|---|
| Stage 1 (Pre-train) | Multimodal | 558K | 558K | - |
| | Text-only | - | - | - |
| Stage 2 (SFT) | Multimodal | 624K | 558K | - |
| | Text-only | 41K | 31K | 10K |

(b) The top-10 multilingual information

| Language | *en* | *zh* | *ko* | *es* | *fr* | *ja* | *de* | *pt* | *zh-tw* | *it* |
|---|---|---|---|---|---|---|---|---|---|---|
| Size | 31K | 3192 | 1219 | 1123 | 1049 | 551 | 435 | 422 | 305 | 234 |

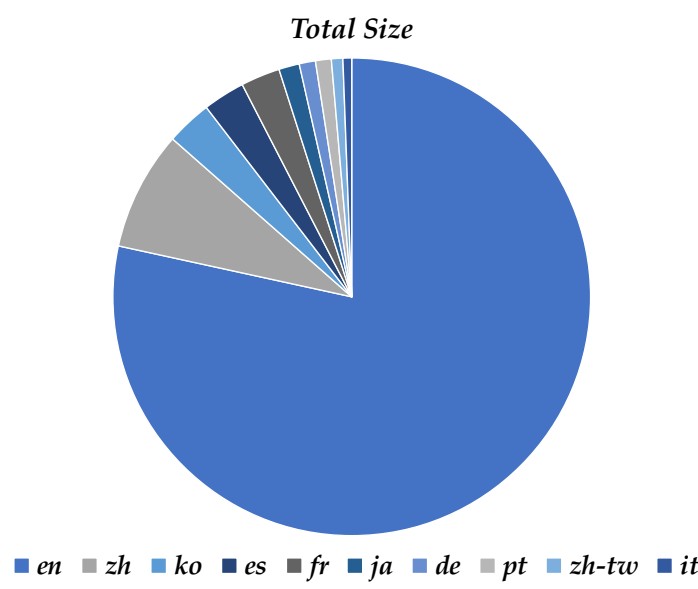

*Total Size*

■ *en* ■ *zh* ■ *ko* ■ *es* ■ *fr* ■ *ja* ■ *de* ■ *pt* ■ *zh-tw* ■ *it*

Figure 8: The pie chart of LLaVA's multilingual data.

# B    RELATED WORK

**Multimodal Large Language Models.** The domain of MLLMs has witnessed significant advances, particularly in the enhancement of visual and language processing. Current MLLMs are usually a combination of visual encoders (Radford et al., 2021; Sun et al., 2023; Fang et al., 2023; Zhang et al., 2022; Oquab et al., 2023; Zhai et al., 2023), LLMs, and fusion modules. Innovations like

Flamingo (Alayrac et al., 2022) have advanced visual representation by integrating a Perceiver Resampler with vision encoders. BLIP-2 (Li et al., 2023b) and InstructBLIP (Dai et al., 2023) employ Q-Former to connect the frozen LLM and vision encoder. InternVL (Chen et al., 2023c) trains huge ViT and QFormer to integrate visual modalities through a multi-stage training method. MiniGPT4 (Zhu et al., 2023) leverages both a Q-Former and a linear projector to bridge the gap between the vision module and LLM. Furthermore, LLaVA (Liu et al., 2023b) adopts a simple MLP projector to promote the alignment between the LLM and vision encoder. mPLUG-Owl (Ye et al., 2023) introduces an approach that begins to finetune the vision encoder and align visual features, followed by tuning the LLM using LoRA (Hu et al., 2021). Qwen-VL (Bai et al., 2023b) improves visual module resolution to 448, aiming to refine the model's visual processing capabilities. Fuyu-8B (Bavishi et al., 2023) directly projects image patches before integration with LLM. MM1 (McKinzie et al., 2024) has conducted ablative studies on connector design choices, revealing that the modality adapter type is less critical than the number of visual tokens and the resolution. MiniGemini (Li et al., 2024c) utilizes high-resolution visual tokens and high-quality data to narrow the performance gap with GPT-4 and Gemini. With the rapid advancements in open-source models, proprietary models such as GPT-4V/4o (OpenAI, 2023b; 2024), Gemini (Team et al., 2023; Reid et al., 2024), Qwen-VL-Plus/MAX (Bai et al., 2023b), and Claude3 (anthropic, 2024) have achieved outstanding results in evaluations and practical applications. In this work, owing to the simplicity of the LLaVA architecture, we adopt a framework similar to LLaVA to design our model.

**Multilingual Multimodal Models.** Recent years have witnessed rapid progress in the expansion of multimodal models to include a wider variety of languages. $M^3P$ (Ni et al., 2021) leverages English as a pivot and alternates between English-only vision-language pre-training and multilingual masked language modeling. In contrast, $UC^2$ (Zhou et al., 2021) translates English captions into various languages and uses images as the anchor. mCLIP (Chen et al., 2023a) enhances the CLIP model by aligning it with a multilingual text encoder through knowledge distillation. Thanks to the expansion of the overall capabilities of large language models (AI, 2024; Bai et al., 2023a; Jiang et al., 2023; Young et al., 2024), their multilingual capacities have significantly improved. Integrating multilingual LLMs with visual abilities has increasingly become a research focus. In the domain of LLMs, PaLI (Chen et al., 2022) develops a 17B multilingual language-image model that spans over 100 languages. Ying-VLM (Li et al., 2023c) discovers that instruction tuning in English can extend its applicability to other languages. Ziya-Visual (Lu et al., 2023) illustrates the translation of English image-text datasets into Chinese, using in-context learning for instruction-response generation. VisCPM (Hu et al., 2023) introduces a training paradigm that fine-tunes the MLLM in a quasi-zero-shot manner based on a strong multilingual large language model. Despite these advancements, they are primarily confined to two languages or rely on the massive translated corpus. On the other hand, there is no suitable multilingual benchmark for MLLMs to evaluate the performance of multiple languages. There are also some multilingual research studies in other domains, such as multilingual machine translation (Zhao et al., 2024; Pires et al., 2023; Purason & Tättar, 2022; Zhang et al., 2021).

## C  TRAINING DETAILS

As shown in Table 4, we provide the training hyperparameters for PARROT. Throughout all stages of training, we consistently train for one epoch, with a batch size of 256 for the first stage and 128 for the second stage. We maintain an image resolution of 336x336 for all two stages and enable the gradient checkpoint mode for each training stage.

## D  EXTENDED EXPERIMENTS

In this section, we further provide more experiments and ablation studies to validate the generality and capability of PARROT across various tasks. Additionally, we present more training details about Figure 1 to offer a clearer understanding for readers.

### D.1  BILINGUAL EVALUATION ON LLAVA-BENCH

VisCPM (Hu et al., 2023) extends the LLaVA-Bench dataset to the Chinese version for bilingual evaluation. To comprehensively compare PARROT with other multilingual models, we conduct experiments on this benchmark. Due to the deprecation of the GPT-4-0314 version by OpenAI, we

Table 4: Training hyperparameters.

| Config | Stage 1 | Stage 2 |
|---|---|---|
| Experts | - | 6 |
| MLP expert network | 2 Linear layers with SiLU | |
| Deepspeed | Zero2 | Zero3 |
| Image resolution | 336×336 | |
| Image encoder | Clip-ViT-L/14-336 | |
| Feature select layer | -2 | |
| Image projector | 2 Linear layers with GeLU | |
| Epoch | 1 | |
| Optimizer | AdamW | |
| Learning rate | 1e-3 | 2e-5 |
| Learning rate scheduler | Cosine | |
| Weight decay | 0.0 | |
| Text max length | 2048 | |
| Batch size per GPU | 16 | 8 |
| GPU | 16 × A100-80G | |
| Precision | Bf16 | |
| Gradient checkpoint | True | |

Table 5: Details on the PARROT's training data, derived from publicly available datasets and our in-house multilingual data.

| Training Stage | Datasets | Samples | Total |
|---|---|---|---|
| Stage 1 | LLaVA-1.5-pretrain (Liu et al., 2023b) | 558K | |
| | Laion-Caption* (Schuhmann et al., 2022) | 12K | 1.2M |
| | CC12M-Caption* (Changpinyo et al., 2021) | 645K | |
| Stage 2 | LLaVA-1.5-finetune (Liu et al., 2023b) | 665K | |
| | ShareGPT4V-zh* (Chen et al., 2023b) | 71K | |
| | ShareGPT4V-pt* (Chen et al., 2023b) | 14K | 793K |
| | ShareGPT4V-ar* (Chen et al., 2023b) | 12K | |
| | ShareGPT4V-tr* (Chen et al., 2023b) | 17K | |
| | ShareGPT4V-ru* (Chen et al., 2023b) | 14K | |

test PARROT in LLaVA-Bench following the version of GPT-4-1106-preview for comparison. As shown in Table 6, PARROT not only demonstrates exceptional ability in the English version of this benchmark but also presents competitive performance in the Chinese version.

Notably, as shown in Table 7, VisCPM uses 140M English data and 1M Chinese data to train the model. In comparison, Qwen-VL-Chat uses 1.1B English data and 300M Chinese data, whereas PARROT only utilizes approximately 2M data in total. Despite using less than 1% of the training data, PARROT achieves remarkable performance in both the English and Chinese versions on LLaVA-Bench. Owing to the architecture we proposed, significant improvement in the model's multilingual capability can be achieved with minimal data usage.

Table 6: Experimental results on LLaVA Test Set accessed by GPT-4. Con: Conversation, DD: Detailed Description, CR: Complex Reasoning, AVG: the average score of three tasks. The best/second best results are marked in **bold** and underlined, respectively. The symbol * denotes that the data are judged following the version of GPT-4-1106-preview because the GPT-4-0314 version is deprecated by OpenAI.

| Model | | LLM Backbone | English | | | | Chinese | | | |
|---|---|---|---|---|---|---|---|---|---|---|
| | | | Con | DD | CR | AVG | Con | DD | CR | AVG |
| English Model | MiniGPT-4 | Vicuna-13B | 65.0 | 67.3 | 76.6 | 69.7 | - | - | - | - |
| | InstructBLIP | Vicuna-13B | 81.9 | 68.0 | 91.2 | 80.5 | - | - | - | - |
| | LLaVA | Vicuna-13B | 89.5 | 70.4 | 96.2 | 85.6 | - | - | - | - |
| En-Zh Bilingual Model | mPLUG-OWL | BLOOMZ-7B | 64.6 | 47.7 | 80.1 | 64.2 | 76.3 | 61.2 | 77.8 | 72.0 |
| | VisualGLM | ChatGLM-6B | 62.4 | 63.0 | 80.6 | 68.7 | 76.6 | 87.8 | 83.6 | 82.7 |
| | Qwen-VL-Chat | Qwen-7B | 82.4 | 76.9 | 91.9 | 83.8 | 82.3 | 93.4 | 89.5 | 88.2 |
| | VisCPM-Balance | CPM-Bee-10B | 75.5 | 64.7 | 91.3 | 77.3 | 85.4 | 81.4 | 96.6 | 88.0 |
| Multilingual Model | PARROT* | Qwen1.5-7B | 82.5 | 71.0 | 89.3 | 81.1 | 82.1 | 88.6 | 92.3 | 87.7 |

Table 7: Comparison of vision encoders, LLMs, and training data in different models.

| Model | vision encoder | LLM | Training Data |
|---|---|---|---|
| mPLUG-Owl | ViT-L/14 (0.3B) | BLOOMZ-7B | - |
| VisualGLM | Q-Former (1.6B) | ChatGLM-6B | English: 300M; Chinese 30M |
| Qwen-VL-Chat | ViT-bigG (1.9B) | Qwen-7B | English: 1.1B; Chinese: 300M |
| VisCPM | Muffin (0.7B) | CPM-Bee-10B | English: 140M; Chinese: 1M |
| PARROT | ViT-L/14 (0.3B) | Qwen1.5-Chat-7B | English: 1.8M; Chinese: 71K |

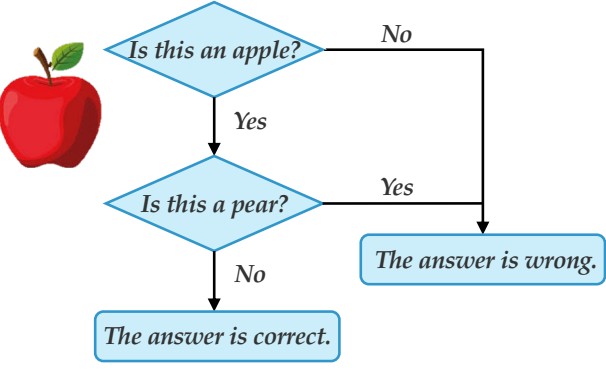

Figure 9: An example of circular evaluation strategy.

## D.2 RADAR CHARTS ON MMBENCH AND MMMB

For a more intuitive demonstration of the multilingual capabilities of PARROT, we present radar charts for the multilingual MMBench and MMMB benchmarks. As depicted in Figure 11a and Figure 11b, our proposed method PARROTexhibits significantly better performance compared to other models.

## D.3 MORE EXPERIMENTAL DETAILS ABOUT DIFFERENT BACKBONES

In this section, we provide detailed information to explain Figure 1. Firstly, to ensure a fair comparison between the OpenAI-CLIP-based model and the Chinese-CLIP-based model, we train distinct models using the same training data as LLaVA, as shown in Table 3a. The hyperparameters are listed in Table 4 without the MoE hyperparameters. As depicted in Figure 1, the OpenAI-CLIP-based model struggles to generate Chinese outputs when given Chinese prompts due to the English-centric training data. In contrast, despite the extremely scarce amount of Chinese training data, the Chinese-CLIP-based model naturally acquires zero-shot capability to understand, process, and generate Chinese texts.

Furthermore, we compare both models on MMBench-CN and MMMB-zh to evaluate their Chinese capability. As shown in Table 9, the performance of the Chinese-CLIP-based model is significantly higher than that of the OpenAI-CLIP-based model. On the other hand, we empirically find that different LLMs have a significant impact on performance. Qwen (Bai et al., 2023a) demonstrates superior Chinese capability compared to Vicuna (Chiang et al., 2023), yet its English capability remains competitive.

## D.4 COMPARISON OF DIFFERENT VISION ENCODERS

We also compare the different vision encoders within the Parrot framework in Table 8. It shows that the Chinese-CLIP-based model maintains comparable multilingual performance to the OpenAI-CLIP-based one. This demonstrates that our framework can be compatible with different vision encoders and achieve multilingual alignment through the MoE module.

Table 8: The comparison of various vision encoders within the PARROT framework.

| Method | LLM | Vision Encoder | MMMB | | | | | | MMBench | | | | | |
| --- | --- | --- | --- | --- | --- | --- | --- | --- | --- | --- | --- | --- | --- | --- |
| | | | en | zh | pt | ar | tr | ru | en | zh | pt | ar | tr | ru |
| LLaVA-1.5 | Vicuna-v1.5-7B | OpenAI-CLIP | 67.07 | 58.83 | 59.76 | 43.50 | 46.43 | 59.06 | 65.37 | 58.33 | 59.02 | 36.16 | 43.90 | 56.95 |
| LLaVA-1.5 | Vicuna-v1.5-7B | Chinese-CLIP | 66.45 | 59.23 | 59.22 | 42.68 | 46.11 | 58.89 | 65.92 | 57.85 | 58.45 | 36.90 | 44.82 | 56.32 |
| ShareGPT4V | Vicuna-v1.5-7B | OpenAI-CLIP | 69.24 | 60.23 | 60.29 | 43.57 | 45.26 | 61.23 | 69.59 | 61.60 | 59.62 | 37.37 | 43.38 | 59.45 |
| ShareGPT4V | Vicuna-v1.5-7B | Chinese-CLIP | 68.65 | 60.85 | 59.49 | 44.33 | 44.90 | 61.88 | 70.28 | 61.91 | 58.83 | 37.00 | 42.55 | 58.97 |
| Parrot | Qwen1.5-7B | OpenAI-CLIP | 70.00 | 68.13 | 67.31 | 62.69 | 58.01 | 66.26 | 70.70 | 70.36 | 65.12 | 57.82 | 58.43 | 64.00 |
| Parrot | Qwen1.5-7B | Chinese-CLIP | 69.22 | 69.24 | 66.32 | 62.15 | 57.77 | 64.31 | 69.95 | 70.87 | 64.92 | 56.57 | 57.13 | 63.15 |

Table 9: The performance of different vision encoders and LLMs on MMBench and MMMB. MMB refers to MMBench. "En/en" represents the English version, and "CN/zh" represents the Chinese version.

| Method | Vision encoder | LLM | MMB-EN | MMB-CN | MMMB-en | MMMB-zh |
| --- | --- | --- | --- | --- | --- | --- |
| LLaVA | OpenAI-CLIP ViT-L/14 | Vicuna 7B | 65.4 | 58.3 | 67.1 | 58.8 |
| LLaVA | OpenAI-CLIP ViT-L/14 | Qwen1.5-Chat 7B | 68.8 | 66.4 | 68.2 | 62.4 |
| LLaVA | Chinese-CLIP ViT-L/14 | Qwen1.5-Chat 7B | 68.1 | 68.3 | 67.6 | 66.1 |
| PARROT | OpenAI-CLIP ViT-L/14 | Qwen1.5-Chat 7B | 70.7 | 70.4 | 70.0 | 68.1 |

## D.5 ABLATION STUDY OF DIFFERENT MULTILINGUAL DATASETS

We conduct an ablation study using only the original LLaVA-1.5-finetune dataset and its translated subsets (∼10K samples for each language) without including ShareGPT4V data in Stage 2. As shown in Table 10, PARROT continues to enhance multilingual performance, confirming the robustness of our framework.

Table 10: The ablation study of the subsets of ShareGPT4V and LLaVA-1.5-finetune.

| Method | Multilingual SFT Dataset | MMMB | | | | | | MMBench | | | | | |
| --- | --- | --- | --- | --- | --- | --- | --- | --- | --- | --- | --- | --- | --- |
| | | en | zh | pt | ar | tr | ru | en | zh | pt | ar | tr | ru |
| Parrot | LLaVA1.5 w/multilingual ShareGPT4V | 70.00 | 68.13 | 67.31 | 62.69 | 58.01 | 66.26 | 70.70 | 70.36 | 65.12 | 57.82 | 58.43 | 64.00 |
| Parrot | LLaVA1.5 w/its multilingual subset | 69.31 | 67.56 | 66.67 | 62.02 | 57.29 | 65.53 | 69.97 | 69.57 | 64.47 | 57.03 | 57.71 | 63.21 |

## D.6 DATA SCALING AND MODEL SIZE SCALING

To further investigate the scaling law in multilingual settings, we have conducted experiments where we progressively expanded the multilingual data (excluding Chinese and English) until it reached a volume comparable to the amount of Chinese data (∼70K). The results, shown in the Table 11, demonstrate that Parrot still satisfies the multilingual scaling law. For instance, the performance on Portuguese improved by 3.0 points, and Arabic saw a gain of 5.2 points. As we increase the multilingual data, the model's performance on the MMMB benchmark continues to improve, suggesting that our model can handle imbalanced multilingual data while still achieving effective scaling and performance gains.

Table 11: The performance comparison on MMMB when using different sample sizes of each language.

| Sample Size (each language) | MMMB | | | | | |
|---|---|---|---|---|---|---|
| | *en* | *zh* | *pt* | *ar* | *tr* | *ru* |
| 10K | 70.0 | 68.1 | 67.3 | 62.7 | 58.0 | 66.3 |
| 30K | 70.1 | 68.0 | 67.6 | 64.1 | 59.9 | 66.7 |
| 50K | 69.9 | 67.9 | 67.8 | 64.8 | 61.4 | 67.2 |
| 70K | 70.3 | 68.4 | 68.3 | 65.7 | 63.2 | 67.4 |

Additionally, we extend Parrot's LLM backbone from Qwen1.5-7B to Qwen1.5-32B, using the same model design and configuration, and evaluate them on the MMMB dataset. As shown in Table 12, the results indicate that Parrot continues to yield better performance even with a larger LLM backbone. This finding validates the idea that the scaling law for model parameters still holds, and our design remains effective as the model size increases. While we are currently limited to the Qwen1.5-32B model, these results suggest that our approach can scale well with model size, and we believe similar trends would be observed with even larger models, such as those with 30B parameters or beyond.

Table 12: The performance comparison on MMMB when using different model sizes of Qwen1.5.

| Method | MMMB | | | | | |
|---|---|---|---|---|---|---|
| | *en* | *zh* | *pt* | *ar* | *tr* | *ru* |
| Parrot-7B | 70.0 | 68.1 | 67.3 | 62.7 | 58.0 | 66.3 |
| Parrot-14B | 73.9 | 71.6 | 69.8 | 68.1 | 64.3 | 70.1 |
| Parrot-32B | 76.3 | 75.4 | 73.8 | 72.1 | 71.2 | 73.5 |

## D.7 COMPARISON WITH LLAVA USING THE SAME DATA

To validate the effectiveness of our proposed approach, we conduct further experiments with an ablation study. Specifically, we expand the baseline LLaVA method by incorporating the same multilingual data used in Parrot. Both models are evaluated on the MMMB dataset, and the results are presented in the Table 13. From the results, we observe that while LLaVA shows a slight improvement with the addition of multilingual data, the increase in performance is limited. In contrast, our Parrot model demonstrates a substantial improvement when multilingual data is included, significantly outperforming LLaVA. This highlights that simply adding multilingual data is not sufficient to bridge the multilingual gap, further emphasizing the effectiveness of our proposed design.

Table 13: We compare the baseline LLaVA with Parrot using the same multilingual training data.

| Method | MMMB | | | | | |
|---|---|---|---|---|---|---|
| | *en* | *zh* | *pt* | *ar* | *tr* | *ru* |
| LLaVA w/o Multilingual data | 67.1 | 58.8 | 59.8 | 43.5 | 46.4 | 59.1 |
| LLaVA w/ Multilingual data | 67.0 | 59.1 | 60.3 | 44.2 | 48.1 | 59.7 |
| Parrot | 70.0 | 68.1 | 67.3 | 62.7 | 58.0 | 66.3 |

## D.8 COMPARISON WITH THE LATEST MODELS

Despite Qwen2-VL and LLaVA-OV being contemporary to our work, we compare to them using the MMMB and multilingual MMBench dataset in Table 14. These models achieve impressive performance as significantly benefiting from significant advancements in LLM backbones and scaling of their datasets. To ensure a fair comparison, we also extend Parrot on top of the Qwen2-7B backbone.

Interestingly, despite Qwen2-VL and LLaVA-OV being trained with over 10x the amount of data used by our model, our Parrot still outperforms them on the multilingual benchmark. This result further demonstrates the effectiveness and robustness of our approach.

Table 14: We extend the Parrot with Qwen2-7B and compare it with the latest models.

| Method | LLM | MMMB | | | | | | MMBench | | | | | |
|---|---|---|---|---|---|---|---|---|---|---|---|---|---|
| | | en | zh | pt | ar | tr | ru | en | zh | pt | ar | tr | ru |
| Qwen2-VL | Qwen2-7B | 80.5 | 80.2 | 78.1 | 74.0 | 71.7 | 79.3 | 79.6 | 79.6 | 75.9 | 71.7 | 70.9 | 76.0 |
| LLaVA-OV | Qwen2-7B | 79.0 | 78.2 | 75.9 | 73.3 | 67.8 | 76.4 | 77.1 | 76.6 | 73.2 | 66.9 | 65.5 | 71.3 |
| Parrot | Qwen1.5-7B | 70.0 | 68.1 | 67.3 | 62.7 | 58.0 | 66.3 | 70.7 | 70.4 | 65.1 | 57.8 | 58.4 | 64.0 |
| Parrot | Qwen2-7B | 80.1 | 80.0 | 79.6 | 76.5 | 75.0 | 79.9 | 78.7 | 78.4 | 76.3 | 75.2 | 74.1 | 77.8 |

## E  FURTHER DESCRIPTION

### E.1  MOE TRAINING STRATEGY

During the first pre-training stage, the MoE module is initialized with random parameters but is not activated or included in the training process. Instead, we focus exclusively on training the projector. This avoids the issue of training a good projector under a randomly initialized MoE. In detail:

1) **Pre-training Stage:** In this stage, the MoE module is bypassed entirely, meaning the image tokens do not pass through the MoE. The primary goal of this stage is to train the projector using a large number of image-text pairs. This enables the projector to align image tokens and textual tokens effectively without interference from the untrained MoE module.

2) **SFT Stage:** Since the SFT stage requires the participation of MoE modules, we randomly initialize the parameters of the MoE components prior to the SFT phase. Once the projector has been trained and achieves robust alignment capabilities in the pre-training stage, we introduce multilingual training data and activate the MoE parameters. At this stage, the MoE is optimized with textual guidance, which drives the alignment of visual tokens while leveraging the well-trained projector. The prior alignment achieved in the pre-training stage allows the MoE to optimize efficiently during this phase.

We present the entire training process of PARROT in the form of pseudocode, as shown in Algorithm 1. It is clear from the algorithm that during the pre-training phase, only the projector is trained. Before the start of the SFT phase, the MoE modules are randomly initialized and incorporated into the training process during the SFT phase.

---

**Algorithm 1** PARROT for MLLM

---

**Input**: Pre-training datasets: $\mathcal{D}^1$, SFT datasets: $\mathcal{D}^2$;

1: Construct the training data's format like LLaVA;
2: Activate the parameters of the projector and freeze others;
3: **for** each data in $\mathcal{D}^1$ **do**                                                    ▷ Pre-training stage
4:     Optimize the projector;
5: **end for**
6: Randomly initialize the parameters of MoE.
7: Activate the parameters of the projector, LLM, and MoE;
8: **for** each data in $\mathcal{D}^2$ **do**                                                    ▷ SFT stage
9:     Select the multilingual experts by the textual guidance;
10:     Optimize the projector, LLM, and MoE;
11: **end for**

---

### E.2  ANALYSIS OF THE TRANSLATION-BASED BASELINE

There is a naive baseline where we first translate the question into English and then translate the English answer back to the target language. On the one hand, our experimental setting follows recent work in multilingual and multimodal large language models (Hu et al., 2023; Zhang et al., 2024a; Hinck et al., 2024), where such a naive baseline has not been commonly considered. While the translation-based approach could be a straightforward alternative, it faces some significant challenges.

First, it is highly susceptible to translation noise, particularly issues related to polysemy and meaning ambiguity between languages. Moreover, our benchmark includes a substantial number of cultural-specific questions, which require deep cultural context knowledge that translation alone cannot effectively capture. In practical use, adding an additional translation step would also introduce extra overhead, increasing both the time and computational cost.

Despite these challenges, we acknowledge the importance of evaluating this baseline and conducting experiments to assess the performance of this translation-based baseline by using the Google Translation API. As shown in the Table 15, the results reveal a "seesaw effect"—while the naive baseline shows some improvements in certain languages, such as Chinese, it leads to performance degradation in others, such as Russian and Portuguese. This highlights the difficulty of addressing multilingualism and multimodal tasks solely through translation.

Table 15: We compare the translation-based baseline with our method. While the naive baseline shows some improvements in certain languages, such as Chinese, it leads to performance degradation in others, such as Russian and Portuguese.

| Method | MMMB | | | | | |
|---|---|---|---|---|---|---|
| | en | zh | pt | ar | tr | ru |
| LLaVA | 67.1 | 58.8 | 59.8 | 43.5 | 46.4 | 59.1 |
| LLaVA w/ translation | 67.1 | 60.7 | 58.6 | 47.3 | 48.6 | 58.9 |
| Parrot | 70.0 | 68.1 | 67.3 | 62.7 | 58.0 | 66.3 |

### E.3 CONSTRUCTION OF THE IN-HOUSE DATASET

Regarding the construction of the dataset, we sample images from the LAION (Schuhmann et al., 2022) and CC12M (Changpinyo et al., 2021) datasets, which encompass a wide variety of categories, including nature, lifestyle, humanities, architecture, cartoons, and abstract art. For each image, we use the Gemini-Pro or GPT-4V API with a unified prompt to generate image descriptions. This prompt ensures that the API generates concise and clear visual information, performs OCR if necessary, and avoids embellishments or subjective interpretations.

Additionally, we generate visual instruction samples from images in the CC12M dataset in a manner similar to ALLaVA (Chen et al., 2024a). We employ Gemini-Pro and GPT-4V to conduct self-questioning and answering tasks, which result in diverse questions and high-quality answers, enriching the dataset further.

In terms of the manual calibration process, our approach indeed follows the same methodology as the MMMB dataset construction. Given that GPT-4 may not perform optimally for certain minor languages (*e.g.*, Arabic and Russian), we introduce a two-stage calibration process to improve performance. This process includes GPT-4 translation followed by manual calibration, as depicted in Figure 3, to address any inaccuracies or biases in the automated generation.

## F BROADER IMPACT AND LIMITATIONS

**Broader Impact.** PARROT leveraging MoE to enhance multilingual alignment presents a positive social impact by promoting linguistic diversity and inclusivity. To address the challenge of the imbalanced language data in SFT datasets and improve non-English visual tokens alignment, this approach contributes to breaking language barriers and facilitating cross-cultural communication, thereby fostering understanding and collaboration across diverse linguistic communities. Additionally, the creation of the Massive Multilingual Multimodal Benchmark (MMMB) fills a crucial gap in evaluating multilingual capabilities, enabling researchers to assess and improve upon models' performance across different languages and cultures. However, it's crucial to acknowledge potential negative social impacts, such as the risk of hallucination. This could potentially result in the model generating misleading or incorrect information, which is a common challenge observed in MLLMs.

**Limitations.** Despite advancements, MLLMs may still exhibit limitations in accurately understanding and responding to complex language-specific contexts, leading to misinformation or misinterpretation

of multilingual inputs. On the other hand, due to the visual component of PARROT being based on CLIP, there are inherent limitations in its ability to process high-resolution images, resulting in the inability to recognize extremely detailed content in some images. Hence, enhancing PARROT's ability to handle high-resolution processing will be part of future work.

## G FUTURE WORK OF MULTILINGUAL BENCHMARK

In future work, we plan to incorporate more culture-related samples in various languages. This will enhance the representation of diverse cultural contexts and ensure that our benchmark accurately reflects the complexities of multilingual interactions. Additionally, we will focus on developing tasks that not only assess linguistic capabilities but also evaluate cultural nuances, which are crucial for effective communication in multilingual settings. By doing so, we aim to provide a more comprehensive evaluation of multilingual models and their performance across different cultural backgrounds.

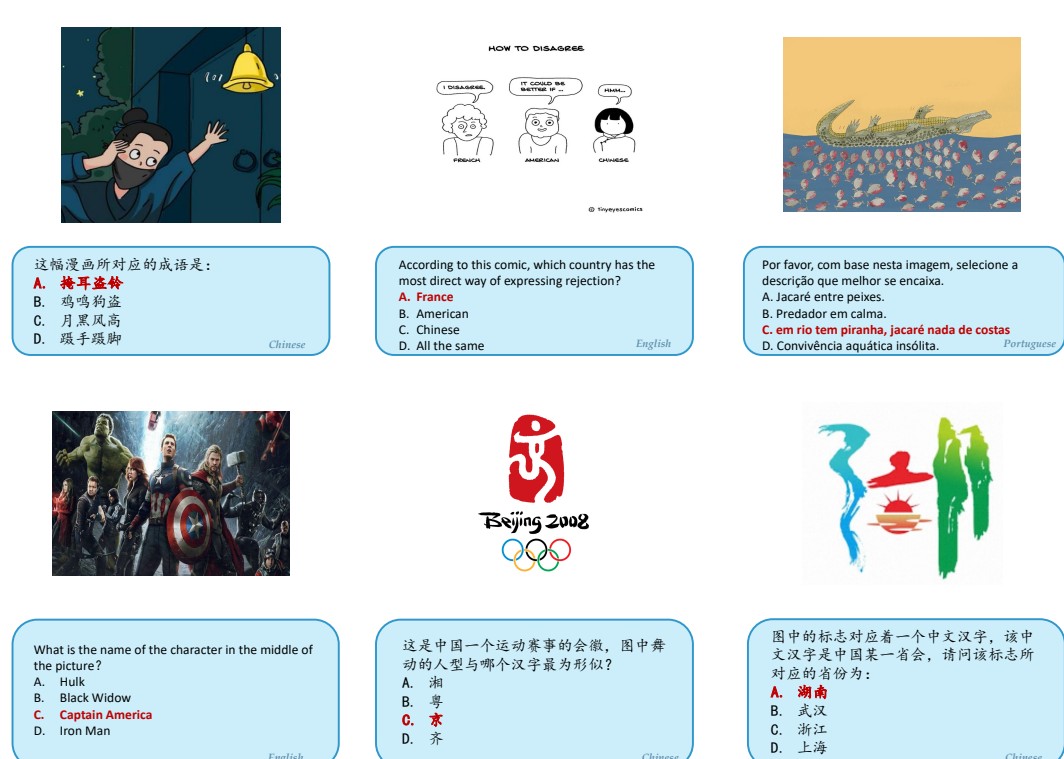

Figure 10: Several culture-related samples in different languages.

## H MORE VISUALIZATION RESULTS

In this section, we include additional visualization results between users' questions and PARROT's responses using multiple languages. These pictures are selected from LLaVA (Liu et al., 2023b) and CuMo (Li et al., 2024b). As depicted in Figures Figures 12 to 17, it is evident that PARROTpossesses superior multilingual capabilities for understanding, processing, and generating multilingual texts. In certain specific cases, PARROT may also experience hallucinations. As depicted in the upper case of Figure 12, it misidentifies Xiaomi SU7 as a Porsche Taycan.

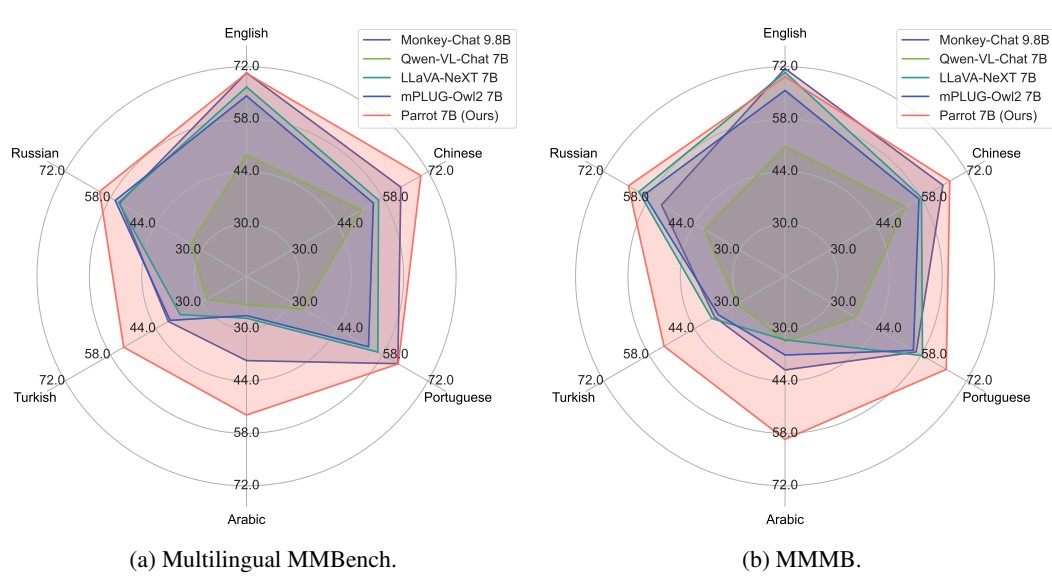

(a) Multilingual MMBench.       (b) MMMB.

Figure 11: The radar charts for the multilingual MMBench and MMMB benchmarks. It demonstrates the impressive multilingual capability of PARROT compared to similar size-based models.

Table 16: **Ablation study on monolingual fine-tuning dataset in MMMB benchmark.** The table shows an effect of performance on six languages when using fine-tuning data from different languages. Models with 7B parameters are used for this ablation.

| Dataset | English | Chinese | Portuguese | Arabic | Turkish | Russian |
|---|---|---|---|---|---|---|
| LLaVA-1.5-finetune | **72.69** | 67.60 | 65.61 | 57.72 | 48.30 | 63.80 |
| + *zh* 71k | 69.18 | **69.06** | 63.92 | 58.13 | 48.95 | 63.63 |
| + *pt* 14k | 69.94 | 68.83 | 65.67 | 58.65 | 51.11 | 63.04 |
| + *ar* 12k | 70.47 | 68.36 | 64.39 | 60.79 | 51.11 | 63.16 |
| + *tr* 17k | 70.82 | 69.01 | 64.85 | 60.76 | **60.70** | 64.39 |
| + *ru* 14k | 69.59 | 68.07 | 64.27 | 60.35 | 53.92 | 64.15 |
| + *zh pt ar tr ru* | 70.00 | 68.13 | **67.31** | **62.69** | 58.01 | **66.26** |

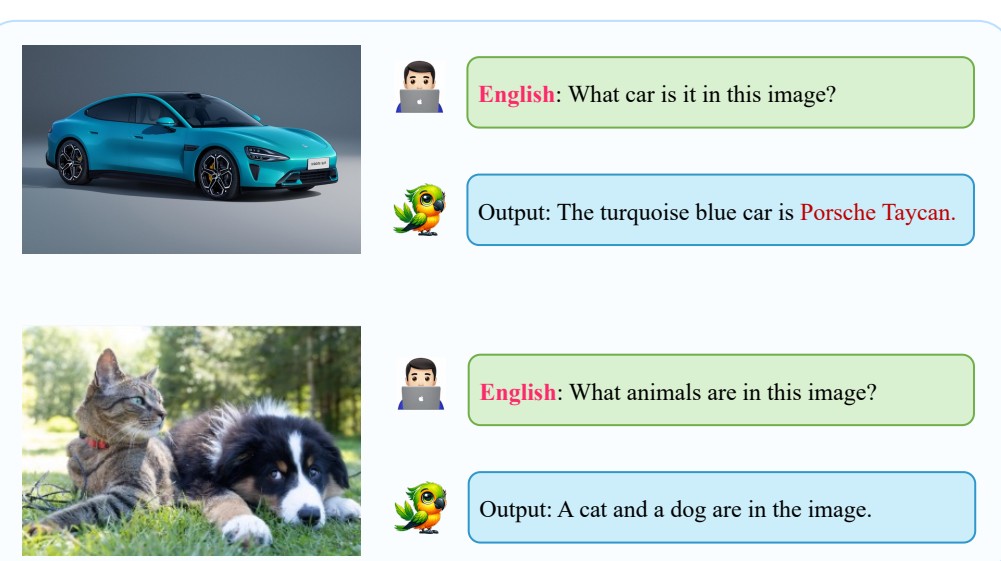

Figure 12: More visualization results between the user and PARROT using English prompts. We highlight the hallucinations from the responses of PARROT.

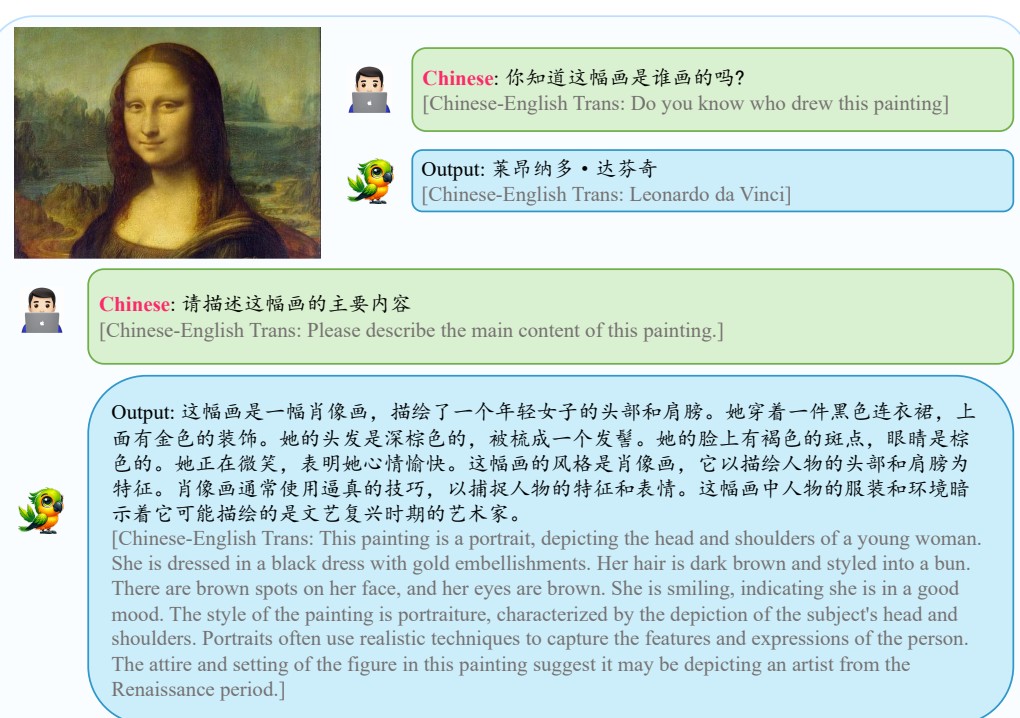

Figure 13: More visualization results between the user and PARROT using Chinese prompts.

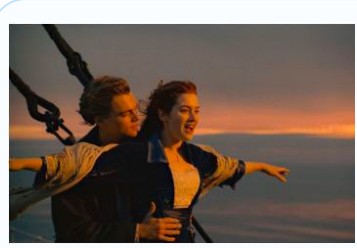

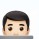
**Portuguese**: Qual é o final deste filme?
[Portuguese-English Trans: What's the ending of this movie?]

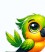
Output: O filme "Titanic" é um romance histórico de 1997 que apresenta uma narrativa fictícia de uma mulher chamada Rose e um homem chamado Jack. A narrativa se passa durante o final do século XIX, quando o navio "Titanic" está se preparando para sua viagem de luxo para Nova York. Jack e Rose se enamoram e se casam, mas a viagem é marcada por um desastre inesperado.
[Portuguese-English Trans: The movie "Titanic" is a 1997 historical romance that features a fictional narrative about a woman named Rose and a man named Jack. The story is set during the late 19th century, as the ship "Titanic" is preparing for its luxury voyage to New York. Jack and Rose fall in love and get married, but the journey is marked by an unexpected disaster.]

Figure 14: More visualization results between the user and PARROT using Portuguese prompts.

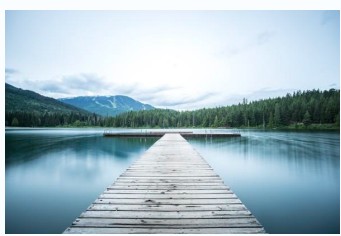

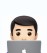
**Arabic**: ما الأشياء التي يجب أن أكون حذرًا منها عند زيارتي هنا؟
[Arabic-English Trans: What's the ending of this movie?]

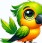
Output: أثناء زيارة هذا الموقع، يجب أن يكون حذرًا من أي مخاطر محتملة، مثل الأسطح المبللة، أو الأشياء المائية، أو أي عوائق محتملة، مثل الصخور أو الصخور أو الأشياء المائية. بالإضافة إلى ذلك، يجب على الزوار أن يكونوا على علم بالعوامل البيئية، مثل تغيرات الطقس، أو درجات الحرارة، أو الظروف المائية، لضمان تجربة آمنة ومرضية.
[Arabic-English Trans: During the visit to this site, one should be cautious of any potential hazards, such as wet surfaces, or water-related objects, or any potential obstacles, like rocks or boulders or aquatic objects. Additionally, visitors should be aware of environmental factors, such as changes in weather, temperature, or water conditions, to ensure a safe and satisfying experience.]

Figure 15: More visualization results between the user and PARROT using Arabic prompts.

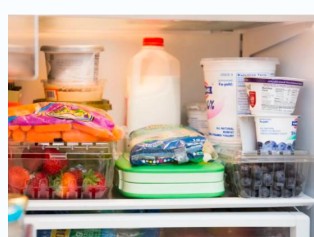
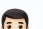

**Turkish**: Bu malzemelerle hangi yemekleri yapabilirim?
[Turkish-English Trans: What are the meals that I can cook with these?]

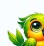

Output: Buzdolabındaki çeşitli malzemelerle, çilekli bir salata, havuçlu bir sandviç, meyveli bir yemek ve süt ve yoğurtla bir yemek gibi çeşitli yemekler yapılabilir. Buzdolabı, havuç, çilek ve yoğurt gibi diğer gıda maddelerini de içerir; bunlar çeşitli yemekler yaratmak için kullanılabilir. Birden fazla yiyecek maddesi ve kapların olması, buzdolabının iyi stoklandığını ve çeşitli yemek seçeneklerine olanak tanıdığını gösterir.
[Turkish-English Trans: With the various items in the refrigerator, one can cook a variety of meals, such as a salad with strawberries, a sandwich with carrots, a meal with berries, and a meal with milk and yogurt. The refrigerator also contains other food items, such as carrots, strawberries, and yogurt, which can be used to create a diverse range of dishes. The presence of multiple food items and containers suggests that the refrigerator is well-stocked, allowing for a variety of meal options.]

Figure 16: More visualization results between the user and PARROT using Turkish prompts.

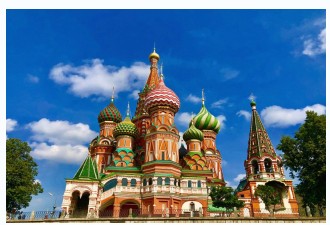
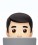

**Russian**: Пожалуйста, помогите мне представить это место на фотографии.
[Russian-English Trans: Please help me introduce the place in this picture.]

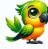

Output: Место, изображенное на картинке, - это собор Василия Блаженного, официально известный как собор Покрова Пресвятой Богородицы на рву. Это знаковое строение находится в Москве, Россия, на Красной площади.
[Russian-English Trans: The place depicted in the picture is the Saint Basil's Cathedral, officially known as the Cathedral of the Intercession of the Most Holy Theotokos on the Moat. This iconic structure is located in Moscow, Russia, on Red Square. ]

Figure 17: More visualization results between the user and PARROT using Russian prompts.

