# OpenReview forum: "Parrot: Multilingual Visual Instruction Tuning"
_ICLR.cc/2025/Conference — Submitted to ICLR 2025_

### Official Review · Reviewer_iiKz · 2024-10-28

**Soundness:** 2
**Presentation:** 3
**Contribution:** 2
**Rating:** 6
**Confidence:** 4

**Summary:**

The paper aims to strengthen the multilingual ability of vision-language models.
Due to the data imbalance problem, vision-language models often perform better on English-based data while suffering from low performance on language with scarce data.

To address the problem, the paper proposes a routing strategy with textual guidance.
The router makes the image embeddings language-aware.
Moreover, the paper collects new data for non-English languages with the assistance of GPT-4.

Experiments on MMBench show reasonable improvements over baseline models, like LLaVA.

**Strengths:**

(1) The paper is clear and easy to follow.
(2) The data imbalance problem in vision-language models is interesting.

**Weaknesses:**

(1) How to train the language transformation experts for the MoE module?
     Are there explicit constraints to optimize the routing?
     As the paper mentions training the baseline LLaVA with the multilingual data suffers from data imbalance problem.
     However, the proposed training strategy can also suffer from the problem:
          I. The routing strategy can be dominated by English-based data.
          II. The language transformation experts should perform worse for languages with fewer data.
     The authors should discuss how the above two problems are addressed.

(2) How does the MoE module affect model performance as the data size of each language increases?
     Also, how does the increased data size affect the baseline model performance, like LLaVA?

(3) The proposed method actually injects language information into the image embeddings, making $H_v$ language-aware.
      Another straightforward baseline is to train several translation expert models, translating other languages into English.

**Questions:**

(1) Carefully study how the performance of language transformation experts affects the overall performance.
(2) Inspect how the router works, like looking into the classification accuracy for different languages.
(3) Compare with the baseline models fairly, like using the same training data.

---

> ### Author Response · Authors · 2024-11-19
> **Response to Reviewer iiKz (1/3)**
>
> Thank you for your kind comments and constructive feedback on our paper, and for appreciating the clarity of our writing and the intriguing exploration of data imbalance in vision-language models.
>
> > **Q1: How to train the language transformation experts for the MoE module? Are there explicit constraints to optimize the routing? As the paper mentions training the baseline LLaVA with the multilingual data suffers from data imbalance problem. However, the proposed training strategy can also suffer from the problem: 1) The routing strategy can be dominated by English-based data. 2) The language transformation experts should perform worse for languages with fewer data.**
>
> A1: Thank you for your insightful comments and constructive feedback. We appreciate the opportunity to clarify our approach, particularly regarding the training of language transformation experts in the MoE module and the potential challenges of data imbalance.
>
> 1. **Training the MoE Module:**
> In our design, we do not include explicit constraints to optimize the router. Instead, the router's behavior emerges based on the guidance of multilingual data. As detailed in **Algorithm 1 in the appendix**, we adopt a two-stage approach for training the MoE module. It is clear from the algorithm that during the pre-training phase, only the projector is trained. This phase is focused on training the projector through a large corpus of image-text pairs, enabling the projector to effectively align the image tokens with the textual tokens. Before the start of the SFT phase, the MoE modules are randomly initialized and incorporated into the training process during the SFT phase.
> In the **SFT stage**, we introduce multilingual training data and activate the MoE parameters for training. During this phase, the routing strategy is dynamically adjusted based on the multilingual text embeddings to select the appropriate experts. This allows us to drive the alignment of multilingual visual tokens using textual guidance. The goal of this stage is to ensure that the model can effectively follow instructions and align visual tokens across different languages.
>
> 2. **Addressing Concerns about the Routing Strategy:**
> We acknowledge the issue of data imbalance, especially with respect to multilingual datasets, which has been a challenge in prior works like LLaVA. However, our proposed training strategy is specifically designed to address this concern. The MoE framework allows for rapid adaptation and specialization of experts across different languages, which helps mitigate the impact of the data imbalance problem. The language transformation experts are trained to perform well across a range of languages, even when the data for some languages is relatively sparse.
> Regarding the concern that the routing strategy could be dominated by English-based data, we want to clarify that our routing strategy is not biased toward English. As shown in **Figure 6c**, we tested the model using Chinese prompts and observed that the MoE experts activated by the input were mostly the ones corresponding to Chinese, with other experts’ logits being relatively small. This demonstrates that our routing mechanism effectively selects the relevant experts based on the language of the input text.
>
> 3. **Performance on Multilingual Benchmarks:**
> Despite the limited amount of multilingual data (e.g., Portuguese and Russian), **Table 1** and **Table 5** show that Parrot performs excellently in these languages. Furthermore, as shown in the table below, **our model's performance improvements in low-resource languages even surpass those in high-resource languages**, indicating that the challenge of MoE underperforming in low-resource languages does not exist in our case.
> The model's robust performance across these low-resource languages highlights the generalization capabilities of our approach and the effectiveness of our routing strategy in dealing with data imbalance.
>
> In conclusion, while data imbalance is an inherent challenge in multilingual learning, our MoE-based approach, combined with a dynamic and language-specific routing strategy, ensures that the model can adapt well to different languages. We believe this approach provides a promising solution for multilingual visual token alignment and avoids the pitfalls of data imbalance.
>
> |Method|LLM|MMMB_en|MMMB_zh|MMMB_pt|MMMB_ar|MMMB_tr|MMMB_ru|
> |-|-|-|-|-|-|-|-|
> |LLaVA1.5|Qwen1.5-7B|67.1|58.8|59.8|43.5|46.4|59.1|
> |Parrot|Qwen1.5-7B|70.0|68.1|67.3|62.7|58.0|66.3|
> |Improvement|-|+2.9|+9.3|+7.5|+19.2|+11.6|+7.2|

---

> > ### Comment · Reviewer_iiKz · 2024-11-25
> > **Thanks for the responses from the authors**
> >
> > Thanks for the detailed feedback from the authors. I still have some concerns.
> > (1)  The authors claim that the router's behavior emerges based on the guidance of multilingual data.
> > From my understanding, the text guidance is to convert CLIP vision features to the word embedding space of a specific language.
> > The converted features could be used for routing, meaning the word embedding space should be discriminative for different languages.
> > The language-specific routing strategy assigns different language transformation experts for different language data. Thus the router could be a classification model for languages. I would like to know the classification accuracy of the router.
> >
> > Moreover, if we could know which experts should be used for both training and inference, What's the model performance on different languages?

---

> > > ### Author Response · Authors · 2024-11-25
> > > **Thank you for your feedback!**
> > >
> > > Thank you for your insightful review and for raising this crucial question!
> > >
> > > First, I would like to clarify that our design takes into account the possibility of different language inputs in a single sentence (e.g. both Chinese and English are in a sentence), so we do not directly assign specific language experts based on the input language. However, since the tasks in our benchmark are designed for the specific languages, **if we were to know the corresponding expert for each language during both training and inference, this would represent the upper bound of our approach.** To assess this, we conduct experiments on the MMMB benchmark. We evaluate the classification accuracy of the router by examining the logits output from the routing process. For each language, we calculate the classification accuracy, as shown in the table below. The results show that classification accuracy is relatively high for Chinese and English but lower for low-resource languages. This indicates that our router performs better on high-resource languages, which aligns with our expectations.
> > >
> > > Additionally, we conduct additional experiments where we provide the specific language expert for each language task during both training and inference. This approach, simulating the upper bound performance, shows significant improvement in languages like Arabic and Turkish, although still lower than the performance on Chinese and English. This suggests that while assigning specific language experts can boost performance, there are still inherent limitations in LLM's performance on low-resource languages.
> > >
> > > We will incorporate these experimental findings into the final version of the paper. In future work, we will explore more efficient routing strategies to further enhance classification accuracy across languages. Thank you again for your feedback!
> > >
> > > |Language|Classification Acc (%)|
> > > |-|-|
> > > |English|92.6|
> > > |Chinese|87.3|
> > > |Portuguese|85.2|
> > > |Arabic|77.6|
> > > |Turkish|74.7|
> > > |Russian|82.6|
> > >
> > > |Methods|Test strategy|MMMB_en|MMMB_zh|MMMB_pt|MMMB_ar|MMMB_tr|MMMB_ru|
> > > |-|-|-|-|-|-|-|-|
> > > |Parrot |Normal|70.0|68.1|67.3|62.7|58.0|66.3|
> > > |Parrot (Upper bound)|Given language expert|70.3|68.5|68.1|65.2|64.6|67.4|

---

> > > > ### Comment · Reviewer_iiKz · 2024-11-25
> > > > **Discussion**
> > > >
> > > > Thanks for the detailed responses from the authors.
> > > > I still have the following questions.
> > > >
> > > > (1) Despite the severe data imbalance problem, the routing model demonstrates a certain degree of invariance to it (best 92.6\% vs. worst 74.7\%). This means that the word embedding space for different languages could be easily distinguished.
> > > > Does this phenomenon correlate to the pre-training of the language models? (Language models pre-trained on multilingual data from scratch should enjoy a unified word embedding space.)
> > > >
> > > > I strongly encourage the authors to validate the effectiveness of the proposed method on various pre-trained language models, like LLaMA, and Vicuna.
> > > >
> > > > (2) Another interesting phenomenon is that:
> > > >      with a 100\% classification accuracy (Upper bound case), the model achieves 68.5\% accuracy on Chinese data. However, with the learned router and an 87.3\% classification accuracy, the model still achieves 68.1\% accuracy on Chinese data. From my understanding, the accuracy is expected to be bounded by 68.5\% * 87.3\%.

---

> > > > > ### Author Response · Authors · 2024-11-25
> > > > > **Thanks for discussion!**
> > > > >
> > > > > Thank you for your valuable and constructive suggestions.
> > > > >
> > > > > **A1:** As outlined in Section 2.2 (Lines 157-159), our MMMB benchmark is specifically designed to evaluate MLLMs across languages with significant differences, ensuring that the benchmark encompasses a wide range of linguistic diversity, as detailed in Figure 4. Therefore, there are inherent linguistic differences between the languages included in the benchmark. Additionally, recent studies [1][2] have shown that models possess "language-agnostic neurons," which align multiple languages in the latent space. These neurons are specifically responsible for language processing, rather than general understanding or reasoning abilities [3][4]. Thus, even with a unified word embedding space, the latent representations for different languages still show some variations. Given the time constraints during the rebuttal discussion stage, we will validate different pre-trained LLMs (e.g., LLaMA and Vicuna) on Parrot as soon as possible and incorporate these results into the final version.
> > > > >
> > > > > [1] How do Large Language Models Handle Multilingualism? NeurIPS 2024.
> > > > > [2] Language-Specific Neurons: The Key to Multilingual Capabilities in Large Language Models. ACL 2024.
> > > > > [3] Unveiling Linguistic Regions in Large Language Models. ACL 2024.
> > > > > [4] Do Llamas Work in English? On the Latent Language of Multilingual Transformers. ACL 2024.
> > > > >
> > > > > **A2:** The classification accuracy is computed based on the largest value in the logits. However, it is important to note that during inference, we do not rely solely on the expert corresponding to the maximum logit value.
> > > > > This phenomenon can be attributed to the specific design of our MoE architecture. In detail, we use a dense MoE, rather than a sparse MoE, as described in Equation 4 (Lines 326-329). Unlike sparse MoE models, which activate a single expert, our approach activates the top-k experts and combines their logits through a weighted softmax (Equation 3). This design allows for the model to leverage multiple experts' outputs, even when there are inherent discrepancies in the largest logits.
> > > > > Due to the cross-linguistic interactions during training, even though the softmax operation may slightly bias the largest logit, the model can still generate effective responses. This phenomenon highlights the robustness of our MoE architecture, which is able to handle multilingual data effectively and maintain high performance even when biases in the logits occur.
> > > > >
> > > > > **If you have any further questions or concerns, please feel free to ask questions anytime. Thank you again for your support in our work!**

---

> > > > > ### Author Response · Authors · 2024-11-27
> > > > > **Further discussion about the different LLM backbones.**
> > > > >
> > > > > We appreciate your suggestions and would like to address the concerns you raised regarding the effectiveness of the proposed method on various pre-trained LLMs.
> > > > >
> > > > > To explore the effectiveness of Parrot on different LLMs, we validate Parrot on both Vicuna-v1.5-7B and LLaMA3-8B backbones and evaluate the performance on the MMMB benchmark, as shown in Table A below. We observe that Parrot performs exceptionally well on the LLaMA3-8B backbone, achieving superior results. This is largely caused by the inherent capabilities of the backbone itself. However, we also find that despite the relatively weaker performance of the Vicuna-v1.5-7B backbone, **Parrot still outperforms the LLaVA-1.5 baseline, especially for low-resource languages such as Arabic (+28.9%) and Turkish (+24.6%).**
> > > > >
> > > > > Furthermore, we explore the router’s classification accuracy across different LLM backbones and notice that the model's performance remains consistent. **It indicates that the model’s characteristics do not significantly change with different LLM backbones.** This suggests that our model exhibits invariance to backbone changes and demonstrates the robustness and transferability of the proposed architecture to different LLMs, confirming the effectiveness of our design. Thank you once again for your detailed feedback.
> > > > >
> > > > >
> > > > > > **Table A: The performance of Parrot with different LLMs on the MMMB benchmark.**
> > > > > |Methods|LLM|MMMB_en|MMMB_zh|MMMB_pt|MMMB_ar|MMMB_tr|MMMB_ru|
> > > > > |-|-|-|-|-|-|-|-|
> > > > > |LLaVA-1.5|Vicuna-v1.5-7B|67.1|58.8|59.8|43.5|46.4|59.1|
> > > > > |Parrot|Qwen1.5-7B|70.0|68.1|67.3|62.7|58.0|66.3|
> > > > > |Parrot|Vicuna-v1.5-7B|68.3|64.7|65.2|56.1|57.8|65.7|
> > > > > |Parrot|LLaMA3-8B|75.6|71.8|71.6|65.3|65.1|68.9|
> > > > >
> > > > > > **Table B: The router's classification accuracy of Parrot with different LLMs.**
> > > > > |Methods|English|Chinese|Portuguese|Arabic|Turkish|Russian|
> > > > > |-|-|-|-|-|-|-|
> > > > > |Parrot w/Qwen1.5-7B|92.6%|87.3%|85.2%|77.6%|74.7%|82.6%|
> > > > > |Parrot w/Vicuna-v1.5-7B|91.7%|86.6%|87.1%|72.8%|75.6%|85.1%|
> > > > > |Parrot w/LLaMA3-8B|94.6%|91.8%|90.5%|81.3%|82.2%|87.6%|
> > > > >
> > > > > > **Table C: The performance of different test strategies.**
> > > > > |Methods|LLM|Test strategy|MMMB_en|MMMB_zh|MMMB_pt|MMMB_ar|MMMB_tr|MMMB_ru|
> > > > > |-|-|-|-|-|-|-|-|-|
> > > > > |Parrot |Vicuna-v1.5-7B|Normal|68.3|64.7|65.2|56.1|57.8|65.7|
> > > > > |Parrot |Vicuna-v1.5-7B|Given language expert|68.8|66.2|66.6|64.1|64.8|66.3|
> > > > > |Parrot |LLaMA3-8B|Normal|75.6|71.8|71.6|65.3|65.1|68.9|
> > > > > |Parrot |LLaMA3-8B|Given language expert|75.8|72.2|72.7|68.9|68.8|70.4|

---

> > > > > > ### Comment · Reviewer_iiKz · 2024-11-27
> > > > > > **Thanks for the feedback from the authors**
> > > > > >
> > > > > > Thanks for your detailed responses.
> > > > > > I'm confused by the results of the LLaVA-1.5 with Vicuna-v1.5-7B. I observe that the LLaVA-1.5 with Vicuna-v1.5-7B achieves the same performance as LLaVA-1.5 with Qwen1.5-7B (provided by the authors in \``Further discussion about the different LLM backbones.\'' and \``Response to Reviewer iiKz (3/3)\'').
> > > > > >
> > > > > > Moreover, what's the baseline performance of LLaVA-1.5 with LLaMA3-8B?

---

> > > > > > > ### Author Response · Authors · 2024-11-27
> > > > > > > **Thanks for the discussion!**
> > > > > > >
> > > > > > > Thank you for your kind comments. First of all, we would like to clarify a potential misunderstanding regarding the performance comparison. The official LLaVA-1.5 implementation does not use the Qwen1.5-7B checkpoint, and in order to ensure consistency with the official version, we have been using LLaVA-1.5 with the Vicuna-v1.5-7B checkpoint for our evaluation. To maintain consistency with the official backbone, all the experiments we conducted previously were based on the Vicuna-v1.5-7B checkpoint. Therefore, the data ablation study (presented in ``Response to Reviewer iiKz (3/3)'') is also based on the Vicuna-v1.5-7B model. **The corresponding results are also shown in Table 1 of the main paper.**
> > > > > > >
> > > > > > > In addition, to address your question about the LLaVA-1.5 with LLaMA3-8B baseline, we evaluate the performance of LLaVA-1.5 with LLaMA3-8B using the open-sourced checkpoint on the MMMB benchmark, as shown in the table below. As observed, LLaVA-1.5 with the LLaMA3-8B backbone performs better than the Vicuna-v1.5-7B-based model by a large margin. However, there remains a noticeable gap in multilingual performance when compared to Parrot  Therefore, we conclude that Parrot demonstrates a clear superiority and effectiveness in these comparisons.
> > > > > > >
> > > > > > > |Methods|LLM|MMMB_en|MMMB_zh|MMMB_pt|MMMB_ar|MMMB_tr|MMMB_ru|
> > > > > > > |-|-|-|-|-|-|-|-|
> > > > > > > |LLaVA-1.5|Vicuna-v1.5-7B|67.1|58.8|59.8|43.5|46.4|59.1|
> > > > > > > |LLaVA-1.5|LLaMA3-8B|74.4|67.5|65.0|58.1|57.7|63.8|
> > > > > > > |Parrot|Vicuna-v1.5-7B|68.3|64.7|65.2|56.1|57.8|65.7|
> > > > > > > |Parrot|LLaMA3-8B|75.6|71.8|71.6|65.3|65.1|68.9|

---

> > > > > > > > ### Comment · Reviewer_iiKz · 2024-11-28
> > > > > > > > **Thanks for the feedback from the authors**
> > > > > > > >
> > > > > > > > Through active discussion with the authors, most of my concerns are addressed.
> > > > > > > > The last point I would like to stress is that the comparisons with previous work should be fair. All the baselines should have the same pre-trained language models and training data. Also, optimal hyper-parameters should be set for each baseline method considering few works are focusing on this problem.

---

> > > > > > > > > ### Author Response · Authors · 2024-11-28
> > > > > > > > > **Thank you for your positive reply!**
> > > > > > > > >
> > > > > > > > > Thank you for raising your score. We are happy we managed to address your concern.
> > > > > > > > >
> > > > > > > > > We greatly appreciate your suggestion and will ensure that all baseline methods are evaluated using the same pre-trained language models and training data. Additionally, we will carefully optimize the hyperparameters for each baseline to ensure a fair comparison and add these experiments in the final version.

---

> ### Author Response · Authors · 2024-11-19
> **Response to Reviewer iiKz (2/3)**
>
> > **Q2: How does the MoE module affect model performance as the data size of each language increases? Also, how does the increased data size affect the baseline model performance, like LLaVA?**
>
> A2: We thank the reviewer for their valuable feedback on the data scaling.
>
> To address the concerns, we conduct additional experiments to explore how the model performance evolves as the amount of multilingual data increases. Specifically, we follow the data scaling methodology outlined in the paper, progressively expanding the multilingual dataset (excluding Chinese and English) to a size comparable to the Chinese dataset (~70K samples). The table below shows the performance of the model on the MMMB dataset as the multilingual data grows. Our findings indicate that Parrot continues to adhere to the multilingual scaling law, with its performance steadily improving as more multilingual data is introduced.
>
> |Multilingual Samples|MMMB_en|MMMB_zh|MMMB_pt|MMMB_ar|MMMB_tr|MMMB_ru|
> |-|-|-|-|-|-|-|
> |10K|70.0|68.1|67.3|62.7|58.0|66.3|
> |30K|70.1|68.0|67.6|64.1|59.9|66.7|
> |50K|69.9|67.9|67.8|64.8|61.4|67.2|
> |70K|70.3|68.4|68.3|65.7|63.2|67.4|
>
> On the other hand, we also investigate the effect of multilingual data scaling on the LLaVA baseline model. During the SFT stage, we incorporate multilingual data into LLaVA and observe a slight improvement in its multilingual capabilities. However, this improvement is quite limited. In contrast, Parrot demonstrates a significant performance boost when multilingual data is added, surpassing LLaVA by a considerable margin. These results suggest that simply adding multilingual data is insufficient to effectively bridge the multilingual gap. This further underscores the effectiveness of the design approach we proposed in our work.
>
> |Methods|MMMB_en|MMMB_zh|MMMB_pt|MMMB_ar|MMMB_tr|MMMB_ru|
> |-|-|-|-|-|-|-|
> |LLaVA w/ 0K|67.1|58.8|59.8|43.5|46.4|59.1|
> |LLaVA w/ 10K|67.0|59.0|60.3|44.1|47.2|59.4|
> |LLaVA w/ 30K|66.8|59.4|60.7|44.6|47.9|59.7|
> |LLaVA w/ 50K|67.1|59.3|61.2|44.4|47.6|60.1|
> |LLaVA w/ 70K|66.7|59.7|61.3|44.8|48.1|60.4|
> |Parrot|70.0|68.1|67.3|62.7|58.0|66.3|
>
> > **Q3: The proposed method actually injects language information into the image embeddings, making Hv language-aware. Another straightforward baseline is to train several translation expert models, translating other languages into English.**
>
> A3: Thank you for your question regarding the naive baseline of translation. We agree that a translation-based approach could be a straightforward alternative. However, it faces some significant challenges.
>
> - **Translation Noise and Ambiguity:**
> A translation-based baseline is inherently sensitive to translation noise, such as errors and ambiguities introduced during translation. For instance, polysemy and context-dependent meanings across languages can lead to inconsistencies that affect the model’s performance.
>
> - **Cultural-Centric Questions in the Benchmark:**
> Our benchmark contains numerous cultural-specific questions that require a deep understanding of cultural knowledge beyond what simple translation can achieve. Such tasks cannot be effectively addressed by merely translating the text into English, as the cultural context is often lost or misinterpreted.
>
> - **Practical Overheads:**
> Adding a translation step introduces additional computational overhead and latency, which can be problematic in real-world applications requiring efficiency. In contrast, our end-to-end design avoids these issues while directly aligning image embeddings with language-specific tokens.
>
> - **Error Propagation with Multiple Expert Models:**
> Training multiple translation expert models to translate other languages into English also introduces the risk of error accumulation, leading to potential instability in model performance. This is particularly challenging in multilingual settings where translation quality can vary significantly between language pairs.
>
> Despite these challenges, we conduct experiments to assess the performance of this translation-based baseline by using the Google Translation API. As shown in the table below, the results reveal a "seesaw effect"——while the naive baseline shows some improvements in certain languages, such as Chinese, it leads to performance degradation in others, such as Russian and Portuguese. This highlights the difficulty of addressing multilingualism and multimodal tasks solely through translation.
>
> We have expanded upon this analysis in Section E.2 of the updated version to further clarify these points. We hope this provides a clearer perspective on the limitations of translation-based approaches in handling multimodal multilingual tasks.
>
> |Methods|MMMB_en|MMMB_zh|MMMB_pt|MMMB_ar|MMMB_tr|MMMB_ru|
> |-|-|-|-|-|-|-|
> |LLaVA|67.1|58.8|59.8|43.5|46.4|59.1|
> |LLaVA w/ translation|67.1|60.7|58.6|47.3|48.6|58.9|
> |Parrot|70.0|68.1|67.3|62.7|58.0|66.3|

---

> > ### Comment · Reviewer_iiKz · 2024-11-25
> > **About data scaling for Parrot and LLaVA**
> >
> > Interestingly, expanding the multilingual dataset (excluding Chinese and English) to a size comparable to the Chinese dataset (~70K samples) has little effect on the performance of the LLaVA models.
> >
> > With the increased data size, I would like to know how the classification accuracy of the router changes.
> > Moreover, as the data size increases, how does the model performance change in different languages if we know which experts should be used for both training and inference?

---

> > > ### Author Response · Authors · 2024-11-25
> > > **Thank you for your feedback!**
> > >
> > > To address this question, we further explore the classification accuracy of the router for different languages as the data size increases. As shown in the table below, with the increase in data samples, the classification accuracy of the router for Arabic and Turkish improves significantly. This suggests that low-resource languages benefit considerably from data scaling, showing substantial gains.
> > >
> > > Furthermore, as the data scaling, the performance of Parrot when given the language expert for specific language tasks also improves. This improvement is particularly noticeable for low-resource languages, while the performance on Chinese and English fluctuates. Therefore, if resources and costs are not taken into account, we could continue expanding the dataset within the Parrot architecture to further enhance model performance.
> > >
> > > |Multilingual Samples|English|Chinese|Portuguese|Arabic|Turkish|Russian|
> > > |-|-|-|-|-|-|-|
> > > |10K|92.6%|87.3%|85.2%|77.6%|74.7%|82.6%|
> > > |30K|92.4%|87.1%|85.6%|78.8%|76.2%|83.1%|
> > > |50K|92.7%|87.4%|86.1%|80.2%|78.7%|83.7%|
> > > |70K|92.3%|87.4%|86.0%|81.6%|80.9%|84.8%|
> > >
> > >
> > > |Methods|Test Strategy|MMMB_en|MMMB_zh|MMMB_pt|MMMB_ar|MMMB_tr|MMMB_ru|
> > > |-|-|-|-|-|-|-|-|
> > > |Parrot 10k|Normal|70.0|68.1|67.3|62.7|58.0|66.3|
> > > |Parrot 10k|Given language expert|70.3|68.5|68.1|65.2|64.6|67.4|
> > > |Parrot 30k|Normal|70.1|68.0|67.6|64.1|59.9|66.7|
> > > |Parrot 30k|Given language expert|70.3|68.4|68.2|66.0|64.8|67.3|
> > > |Parrot 50k|Normal|69.9|67.9|67.8|64.8|61.4|67.2|
> > > |Parrot 50k|Given language expert|**70.4**|68.5|**68.5**|66.2|64.9|68.1|
> > > |Parrot 70k|Normal|70.3|68.4|68.3|65.7|63.2|67.4|
> > > |Parrot 70k|Given language expert|70.2|**68.7**|**68.5**|**66.4**|**65.2**|**68.3**|

---

> ### Author Response · Authors · 2024-11-19
> **Response to Reviewer iiKz (3/3)**
>
> > **Q4: Compare with the baseline models fairly, like using the same training data.**
>
> A4: Thank you for your valuable feedback regarding the fairness of model comparisons. We understand the importance of using the same training data to ensure a fair comparison and have taken steps to address this concern.
>
> To validate the effectiveness of our proposed approach, we conduct further experiments with an ablation study. Specifically, we expand the baseline LLaVA method by incorporating the same multilingual data used in Parrot. Both models are evaluated on the MMMB dataset, and the results are presented in the table below. From the results, we observe that while LLaVA shows a slight improvement with the addition of multilingual data, the increase in performance is limited. In contrast, our Parrot model demonstrates a substantial improvement when multilingual data is included, significantly outperforming LLaVA. This highlights that simply adding multilingual data is not sufficient to bridge the multilingual gap, further emphasizing the effectiveness of our proposed design.
>
> Moreover, the findings from the ablation study in **Figure 6a of the main paper** further support this conclusion, reinforcing the validity of our design.
>
> |Methods|MMMB_en|MMMB_zh|MMMB_pt|MMMB_ar|MMMB_tr|MMMB_ru|
> |-|-|-|-|-|-|-|
> |LLaVA w/o Multilingual data|67.1|58.8|59.8|43.5|46.4|59.1|
> |LLaVA w/ Multilingual data|67.0|59.1|60.3|44.2|48.1|59.7|
> |Parrot|70.0|68.1|67.3|62.7|58.0|66.3|

---

### Official Review · Reviewer_cBkD · 2024-11-02

**Soundness:** 3
**Presentation:** 3
**Contribution:** 3
**Rating:** 6
**Confidence:** 3

**Summary:**

This paper addresses the imbalance in the quantity of SFT data for different languages within training datasets used in MLLM's SFT process, which results in suboptimal alignment performance for various languages with limited data. To tackle this issue, the authors propose a novel structure that combines the cross-attention and MoE structure, enabling the input visual tokens to MLLM to be conditioned on the language input. Furthermore, to address the current lack of comprehensive benchmarks for multilingual multimodal tasks, the authors propose a new benchmark, named MMMB, which provides a more extensive evaluation for multilingual MLLMs. Extensive experiments validate the effectiveness of the proposed method on multilingual benchmarks.

**Strengths:**

- The author's observation regarding the lack of balance among different languages in the SFT data of MLLMs is insightful.
- The proposed method is simple and efficient.
- The benchmark proposed by the author is rigorous and highly relevant for evaluating multilingual MLLMs.
- The paper is well-written and easy to follow.
- The experiments are comprehensive, demonstrating the effectiveness of the method proposed by the author.

**Weaknesses:**

- **Regarding the issue of alignment:** I agree with the author's observation that the imbalance of SFT data among different languages may lead to poor alignment between vision tokens and multilingual tokens in MLLMs. However, it should be noted that the alignment data in the pretraining phase consists of English-only data, and the amount of data in the pretraining phase is significantly larger than that in the SFT phase. Would the impact of alignment between visual tokens and different language tokens be more severe in the pretraining phase?
- **Lack of comparison for the latest MLLM models**: Some of the VLMs the author compares are outdated. Could the evaluation include the latest VLMs, such as Qwen2-VL [1] and LLaVA-OV [2]?
- **About the scalibility**: Introducing a language-aware structure is an effective approach, but if similar structures are not introduced, would simply increasing the proportion of different language data in the SFT data yield a similar improvement in model performance? In larger models, such as those with 30B or larger model, is the performance gain from this model design consistent?

[1] Peng Wang, et al. "Qwen2-VL: Enhancing Vision-Language Model's Perception of the World at Any Resolution" arXiv preprint arXiv:2409.12191 (2024)

[2] Bo Li, et al. "LLaVA-OneVision: Easy Visual Task Transfer" arXiv preprint arXiv:2408.03326 (2024)

**Questions:**

Please see the above weaknesses.

---

> ### Author Response · Authors · 2024-11-19
> **Thank you for your detailed, positive, and encouraging review (1/2)**
>
> Thank you for your insightful comments and for appreciating our insightful observation, the simplicity and efficiency of our method, the rigor of our benchmark, and the clarity and comprehensiveness of our experiments.
>
> > **Q1: I agree with the author's observation that the imbalance of SFT data among different languages may lead to poor alignment between vision tokens and multilingual tokens in MLLMs. However, it should be noted that the alignment data in the pretraining phase consists of English-only data, and the amount of data in the pretraining phase is significantly larger than that in the SFT phase. Would the impact of alignment between visual tokens and different language tokens be more severe in the pretraining phase?**
>
> A1: Thank you for your thoughtful observation. We would like to address the concern you raised regarding the potential impact of alignment during the pretraining phase, given that the alignment data is predominantly in English and the pretraining dataset is much larger than the SFT dataset.
>
> We acknowledge that the pre-training phase involves significantly larger amounts of data compared to the SFT phase, and incorporating multilingual data during pre-training could enhance alignment to some extent. **However, in practice, it is challenging to collect sufficient high-quality multilingual image-text pairs at the scale required for pre-training.** This limitation is a key factor that influenced our design choices, underscoring the importance of using image-text pairs to align visual and textual features through training the projector. The detailed training strategy is outlined below:
>
> 1. During the pre-training phase, we leverage a large number of coarse-grained image-text pairs to train the projector, aligning the visual and textual tokens. The focus is exclusively on refining the projector’s capability to produce closely aligned hidden states for these tokens. Importantly, the parameters of the LLM are not updated in this phase, meaning the multilingual abilities of the LLM remain unaffected. This ensures that no degradation of multilingual capabilities occurs despite the use of English-only data.
>
> 2. In contrast, during the SFT phase, we incorporate multilingual training data during the SFT phase. In this stage, the MoE parameters are activated and trained alongside the model. The textual guidance provided by the multilingual data further enhances the alignment of visual tokens with multilingual textual tokens, enabling the model to effectively strengthen its multilingual alignment capabilities while also gaining instruction-following skills.
>
> In summary, while the pre-training phase helps to align visual and textual tokens, **it is the SFT phase where we see the most significant improvements in multilingual alignment**, especially due to the inclusion of diverse language data and the active participation of MoE parameters in training.
>
> > **Q2: Some of the VLMs the author compares are outdated. Could the evaluation include the latest VLMs, such as Qwen2-VL and LLaVA-OV?**
>
> A2: Thank you for your valuable feedback. Despite Qwen2-VL and LLaVA-OV being contemporary to our work, we compare to them on the MMMB and multilingual MMBench dataset in the table below. These models achieve impressive performance as significantly benefiting from significant advancements in LLM backbones and scaling of their datasets. To ensure a fair comparison, we also extend Parrot on top of the Qwen2-7B backbone.
>
> Interestingly, despite Qwen2-VL and LLaVA-OV being trained with over 10x the amount of data used by our model, our Parrot still outperforms them on the multilingual benchmark. This result further demonstrates the effectiveness and robustness of our approach. **In this revision, we have annotated the final performance of each method in Table 14.**
>
> |Method|LLM|MMMB_en|MMMB_zh|MMMB_pt|MMMB_ar|MMMB_tr|MMMB_ru|
> |-|-|-|-|-|-|-|-|
> |Qwen2-VL|Qwen2-7B|80.5|80.2|78.1|74.0|71.7|79.3|
> |LLaVA-OV|Qwen2-7B|79.0|78.2|75.9|73.3|67.8|76.4|
> |Parrot|Qwen1.5-7B|70.0|68.1|67.3|62.7|58.0|66.3|
> |Parrot|Qwen2-7B|80.1|80.0|79.6|76.5|75.0|79.9|
>
> |Method|LLM|MMB_en|MMB_zh|MMB_pt|MMB_ar|MMB_tr|MMB_ru|
> |-|-|-|-|-|-|-|-|
> |Qwen2-VL|Qwen2-7B|79.6|79.6|75.9|71.7|70.9|76.0|
> |LLaVA-OV|Qwen2-7B|77.1|76.6|73.2|66.9|65.5|71.3|
> |Parrot|Qwen1.5-7B|70.7|70.4|65.1|57.8|58.4|64.0|
> |Parrot|Qwen2-7B|78.7|78.4|76.3|75.2|74.1|77.8|

---

> ### Author Response · Authors · 2024-11-19
> **Thank you for your detailed, positive, and encouraging review (2/2)**
>
> > **Q3: Introducing a language-aware structure is an effective approach, but if similar structures are not introduced, would simply increasing the proportion of different language data in the SFT data yield a similar improvement in model performance?**
>
> A3: Thank you for your valuable feedback. To address this, we conduct an ablation experiment to further validate the effectiveness of our proposed approach. Specifically, we expand the baseline LLaVA method by incorporating the same multilingual data as used in our Parrot model and evaluate the performance on the MMMB dataset. As shown in the table below, we observe that while adding multilingual data to LLaVA results in a modest improvement, the gain is limited. This suggests that while increasing the amount of multilingual data can help, the baseline LLaVA model still struggles to effectively align visual and textual tokens at the multilingual level. Without a dedicated mechanism for managing linguistic diversity and guiding alignment, the performance improvements plateau.
>
> In contrast, Parrot shows a substantial increase in performance when multilingual data is added, outperforming LLaVA by a significant margin. This highlights that merely adding more multilingual data does not sufficiently bridge the multilingual gap. Instead, it is the combination of our language-aware structure with multilingual data that enables the substantial performance boost observed in our approach. Moreover, the findings from the ablation study in **Figure 6a of the main paper** further support this conclusion, reinforcing the validity of our design.
>
> Additionally, we refer to the results in **Table 6 and Table 7 of the appendix**, where we observe that an increase in the volume of data does not necessarily lead to superior multilingual performance. For instance, models like VisCPM, mPLUG-Owl, and Qwen-VL have utilized a very large amount of Chinese and English data (100M+), but they do not exhibit significant advantages over the Parrot model, which leverages a more carefully designed architecture and the slight multilingual data. This suggests that while data quantity plays an important role, the quality of the data and the architecture design are also crucial factors for achieving robust multilingual capabilities.
>
> |Methods|MMMB_en|MMMB_zh|MMMB_pt|MMMB_ar|MMMB_tr|MMMB_ru|
> |-|-|-|-|-|-|-|
> |LLaVA w/o Multilingual data|67.1|58.8|59.8|43.5|46.4|59.1|
> |LLaVA w/ Multilingual data|67.0|59.1|60.3|44.2|48.1|59.7|
> |Parrot|70.0|68.1|67.3|62.7|58.0|66.3|
>
> > **Q4: In larger models, such as those with 30B or larger model, is the performance gain from this model design consistent?**
>
> A4: Due to resource and time constraints, we extend Parrot's LLM backbone from Qwen1.5-7B to Qwen1.5-32B, using the same model design and configuration, and evaluate them on the MMMB dataset. The results indicate that Parrot continues to yield better performance even with a larger LLM backbone. This finding validates the idea that the scaling law for model parameters still holds, and our design remains effective as the model size increases.
>
> While we are currently limited to the Qwen1.5-32B model, these results suggest that our approach can scale well with model size, and we believe similar trends would be observed with even larger models, such as those with 30B parameters or beyond.
>
> |Method|MMMB_en|MMMB_zh|MMMB_pt|MMMB_ar|MMMB_tr|MMMB_ru|
> |-|-|-|-|-|-|-|
> |Parrot-7B|70.0|68.1|67.3|62.7|58.0|66.3|
> |Parrot-14B|73.9|71.6|69.8|68.1|64.3|70.1|
> |Parrot-32B|76.3|75.4|73.8|72.1|71.2|73.5|

---

> ### Comment · Reviewer_cBkD · 2024-11-25
> **Thanks for the responses from the authors**
>
> Thanks for your detailed responses, my main concern has been addressed. I think the author's response was quite good, and I will keep my score. I would recommend accepting this paper.

---

> > ### Author Response · Authors · 2024-11-25
> > **Thanks for the feedback!**
> >
> > We are glad that we have addressed your concerns. **If possible, we would greatly appreciate any positive feedback or a potential increase in your rating.** If you have any further questions or concerns, **please feel free to ask questions anytime.** Thank you again for your support in our work!

---

### Official Review · Reviewer_F1M5 · 2024-11-04

**Soundness:** 3
**Presentation:** 3
**Contribution:** 2
**Rating:** 5
**Confidence:** 4

**Summary:**

This paper proposes Parrot, an MLLM designed to handle multilingual tasks. Parrot follows the LLava architecture, introducing an additional MoE module after the visual projector to enhance multilingual understanding. During training, Parrot translates public datasets into multiple languages and adopts a two-stage training scheme similar to LLava. To assess multilingual capabilities in MLLMs, the paper introduces MMMB (Massive Multilingual Multimodal Benchmark), encompassing 6 languages, 15 categories, and 12,000 questions. Parrot demonstrates strong performance on both MMBench and MMMB.

**Strengths:**

1. High reusability—the proposed approach is simple and highly portable.
2. Extensive experimentation—the paper validates multilingual capability across multiple benchmarks.
3. Open-sourced multilingual benchmark dataset (MMMB) to support multilingual evaluation for MLLMs.

**Weaknesses:**

1. The in-house dataset is not discussed in detail, such as whether it will be open-sourced, and whether the manual calibration process mentioned in lines 365-366 follows the same methodology as MMMB construction.
2. There is no experimental analysis on Parrot's performance loss in a single language, for example, whether the use of multilingual data and the MoE module reduces the model's English proficiency.

**Questions:**

1. As mentioned in Weakness #1, will the in-house dataset be open-sourced? The construction process should be explained in more detail, particularly regarding noise control and data diversity.
2. As mentioned in Weakness #2.
3. Parrot’s training consists of two stages, largely following LLava’s approach. However, the inclusion of the MoE architecture raises questions about its integration in stage 1. Specifically, how are the MoE weights initialized? If initialized randomly, is it optimal to include the MoE in stage 1, given that this stage focuses on aligning multimodal features? Additionally, if the MoE is indeed included in stage 1, an ablation study on whether to freeze or not freeze the MoE module would be insightful.

---

> ### Author Response · Authors · 2024-11-19
> **Response to Reviewer F1M5 (1/2)**
>
> Thank you for your kind comments and constructive feedback on our paper, and for appreciating the high reusability, extensive experiments, and open-sourced MMMB that support multilingual evaluation for MLLMs.
>
> > **Q1: The in-house dataset is not discussed in detail, such as whether it will be open-sourced, and whether the manual calibration process mentioned in lines 365-366 follows the same methodology as MMMB construction. Will the in-house dataset be open-sourced? The construction process should be explained in more detail, particularly regarding noise control and data diversity.**
>
> A1: Thank you for your valuable comments regarding the in-house dataset and its construction process. To address these concerns, we have made the Parrot code and dataset publicly available on an anonymous GitHub repository (**[Code and Dataset](https://anonymous.4open.science/r/Parrot-Anonymous-FDC2)**), aiming to facilitate further research and engagement from the community.
>
> Regarding the construction of the dataset, we sample images from the LAION [1] and CC12M [2] datasets, which encompass a wide variety of categories, including nature, lifestyle, humanities, architecture, cartoons, and abstract art. For each image, we use the Gemini-Pro or GPT-4V API with a unified prompt to generate image descriptions. This prompt ensures that the API generates concise and clear visual information, performs OCR if necessary, and avoids embellishments or subjective interpretations.
>
> Additionally, we generate visual instruction samples from images in the CC12M dataset in a manner similar to ALLaVA [3]. We employ Gemini-Pro and GPT-4V to conduct self-questioning and answering tasks, which result in diverse questions and high-quality answers, enriching the dataset further.
>
> In terms of the manual calibration process, our approach indeed follows the same methodology as the MMMB dataset construction. Given that GPT-4 may not perform optimally for certain minor languages (e.g., Arabic and Russian), we introduce a two-stage calibration process to improve performance. This process includes GPT-4 translation followed by manual calibration, as depicted in **Figure 3 of the main paper**, to address any inaccuracies or biases in the automated generation. In detail, we begin by using GPT-4 to translate the original problem into the target language. Then, we input the first translation result back into GPT-4 for a re-check and refinement. This step helps to identify and correct any immediate errors or inconsistencies in the translation. For manual calibration, we engage two groups of professional translators for each language involved in the study:
>
> **First Group - Refinement**: Each group consists of three language experts who independently review and refine the translations produced by GPT-4. This step results in three distinct translation versions for each piece of content.
> **Second Group - Voting**: The second group of experts is responsible for evaluating these three refined translations. Through a voting process, they select the best translation that accurately captures the intended meaning and nuances of the original text.
>
> **In this revision, we have supplied further clarification about the detailed construction of our in-house dataset in Section E.3.**
>
> [1] LAION-5B: An open large-scale dataset for training next-generation image-text models. NeurIPS 2022
> [2] Conceptual 12M: Pushing web-scale image-text pre-training to recognize long-tail visual concepts. CVPR 2021
> [3] ALLaVA: Harnessing GPT-4V synthesized data for a lite vision-language model. Arxiv 2024

---

> ### Author Response · Authors · 2024-11-19
> **Response to Reviewer F1M5 (2/2)**
>
> > **Q2: There is no experimental analysis on Parrot's performance loss in a single language, for example, whether the use of multilingual data and the MoE module reduces the model's English proficiency.**
>
> A2: Thank you for your valuable feedback and the opportunity to clarify the concerns regarding Parrot's performance in a single language. To analyze this, we have conducted a series of ablation experiments, which are detailed in **Appendix Table 16** and the table below. Additionally, we have conducted the MoE ablation experiment in Figure 6a, which shows a significant improvement in each language, demonstrating the robustness and effectiveness of the MoE module.
>
> **Monolingual Dataset Analysis:**
> When fine-tuning on a single language, we observe a slight decrease in English proficiency. However, cross-linguistic interactions often provide positive effects. For example, adding Portuguese data led to notable improvements in Chinese and Turkish performance, with scores increasing from 67.60 to 68.83 for Chinese and from 48.30 to 51.11 for Turkish. This suggests that certain multilingual datasets enhance model robustness across languages without significantly impairing English capabilities.
>
> **Multilingual Dataset Impact (Table 2):**
> Experiments on the multilingual MMBench reveal that incorporating multilingual data improved English performance from 69.4 to 70.7. This indicates that the inclusion of multilingual data does not inherently degrade English proficiency but results in minor, context-dependent fluctuations.
>
> **MoE Module Ablation (Figure 6a):**
> The MoE ablation study demonstrates significant improvements across all tested languages, including English. This underscores the MoE module's effectiveness in leveraging multilingual data to enhance language-specific capabilities while maintaining robustness.
>
> In conclusion, while minor variations in English proficiency may occur during specific single-language fine-tuning, the overall results show that multilingual data and the MoE module contribute positively to model performance across languages, including English.
>
> |Dataset|MMMB_en|MMMB_zh|MMMB_pt|MMMB_ar|MMMB_tr|MMMB_ru|
> |-|-|-|-|-|-|-|
> |LLaVA-1.5-finetune|**72.69**|67.60|65.61|57.72|48.30|63.80|
> |+ zh 71K|69.18|**69.06**|63.92|58.13|48.95|63.63|
> |+ pt 14K|69.94|68.83|65.67|58.65|51.11|63.04|
> |+ ar 12K|70.47|68.36|64.39|60.79|51.11|63.16|
> |+ tr 17K|70.82|69.01|64.85|60.76|**60.70** |64.39|
> |+ ru 14K|69.59|68.07|64.27|60.35|53.92|64.15|
> |+ zh pt ar tr ru|70.00|68.13|**67.31**|**62.69**|58.01|**66.26**|
>
> > **Q3: Parrot’s training consists of two stages, largely following LLava’s approach. However, the inclusion of the MoE architecture raises questions about its integration in stage 1. Specifically, how are the MoE weights initialized? If initialized randomly, is it optimal to include the MoE in stage 1, given that this stage focuses on aligning multimodal features? Additionally, if the MoE is indeed included in stage 1, an ablation study on whether to freeze or not freeze the MoE module would be insightful.**
>
> A3: Thank you for your insightful review and comments.
>
> Regarding your concern about the MoE module and its role during different stages of training, we would like to clarify that in the pre-training stage, the MoE module is randomly initialized and effectively skipped. In this stage, the main objective is to train the projector using a large number of image-text pairs, enabling the projector to align image tokens and textual tokens effectively. Since the visual tokens do not pass through the MoE module during pre-training, the projector can focus solely on learning this alignment.
>
> In the subsequent SFT stage, we introduce multilingual training data and activate the MoE parameters for training. At this stage, our goal is to enhance the model's instruction-following capabilities while also leveraging textual guidance to drive visual token alignment. Because the projector has already learned a strong alignment capability during the pre-training stage, it can now work with the MoE module to rapidly optimize visual token alignment. **We present the entire training process of Parrot in the form of pseudocode, as shown in Algorithm 1 in the appendix**. It is clear from the algorithm that during the pre-training phase, only the projector is trained. Before the start of the SFT phase, the MoE modules are randomly initialized and incorporated into the training process during the SFT phase.
>
> In conclusion, the MoE module is not activated or included in the pre-training stage, while we focus exclusively on training the projector. **In this revision, we have supplied further clarification about the MoE training strategy in Section E.1.** This clarification will help to highlight how the MoE module interacts with the projector at each stage and how this contributes to the model’s overall multilingual proficiency.

---

> ### Author Response · Authors · 2024-11-25
> **Appreciating Your Reviews and Humbly Asking for Feedback**
>
> Dear Reviewer F1M5,
>
> We sincerely appreciate your great efforts in reviewing this paper.
>
> During the remaining hours of the author-reviewer discussion period, it would be great if you could inform us whether our response has addressed your concerns regarding our paper. Your dedication to reviewing our work despite your busy schedule is genuinely appreciated. Lastly, we just want to say thank you for your evaluation of both our paper and our rebuttal.
>
> Best regards,
>
> Authors of paper 5775

---

> ### Author Response · Authors · 2024-11-28
> **Please let us know if you have any further questions**
>
> Dear Reviewer F1M5,
>
> We express our gratitude for the time and effort you have dedicated as a reviewer for ICLR 2025. We hope the revisions have addressed your concerns and if you have any remaining questions or further concerns, please feel free to ask us anytime. We will continue to work hard to make our work better and try to address any further questions before the discussion period ends. Wishing you a happy Thanksgiving!
>
> With warm regards,
>
> Authors of paper 5775

---

### Official Review · Reviewer_foLk · 2024-11-04

**Soundness:** 3
**Presentation:** 4
**Contribution:** 4
**Rating:** 6
**Confidence:** 4

**Summary:**

This paper aims at the multilingual issues in recent MLLMs and proposes a MoE-based alignment layer to this end. First, the authors find that existing MLLMs are not friendly to non-English queries and analyze this from the imbalanced training datasets. A simple solution would be to train a new adapter for each language, but this approach is not feasible due to the limited number of training pairs. This paper thus introduces Parrot, which trains a soft adapter under the MoE framework. To fully test the multilingual ability of MLLMs, this paper also releases a Massive Multilingual Multimodal Benchmark(MMMB). Results on MMMB and MMBench show that Parrot has a better multilingual alignment with limited training data.

**Strengths:**

1) The problem that improves the multilingual abilities of MLLMs is an open and challenging problem. This paper develops a simple and efficient insight for the community.

2) This idea that uses MoE-based adapter is interesting. It can learn from the limited image-text pairs and show good performance empirically.

3) The collected new benchmark MMMB would be useful for subsequent research.

4) Extensive comparison and ablations show the efficiency of the proposed model.

**Weaknesses:**

1) Parrot  depends on balanced datasets and does not consider unbalanced situations, which are the most common in practice. That is saying that Parrot may not satisfy the scaling law. When considering massive training pairs, unbalanced cases occur, which could lead to sub-optimal MoE learning, sticking in the same predicament as existing MLLMs.

2) Lack of a strong baseline. I wonder about the performance of a naive baseline where we first translate the question into English and then translate the English answer back to the target language.

**Questions:**

1) At the first pretraining stage, is the MoE initialized with random parameters? If yes, how can we learn a good Projector under a random MoE?

2) An open question: What is the most real benefit for today's MLLMs? Recent MLLMs can be divided into two groups: 1) dataset-driven, these models adopt simple adaptor to map images into the language space (Qwen-vl, llava, gpt-4o, gemini Pro) and jointly train MLLMs on massive image-text data. 2) tokenization-based models, these models believe a good image tokenization can align the image and text well (Parrot, [1][2]). In my opinion, the simple structure and target datasets fine-tuning may have better robustness and improvement than small structure-based modifications.

[1] https://arxiv.org/abs/2408.05019
[2] https://arxiv.org/pdf/2405.01926

---

> ### Author Response · Authors · 2024-11-19
> **Thank you for your detailed, positive, and encouraging review (1/3)**
>
> We extend sincere gratitude to the reviewer for their insightful comments and for greatly appreciating our extensive experiments, interesting ideas, useful benchmark, and efficient insight.
>
> > **Q1: Parrot depends on balanced datasets and does not consider unbalanced situations, which are the most common in practice. That is saying that Parrot may not satisfy the scaling law. When considering massive training pairs, unbalanced cases occur, which could lead to sub-optimal MoE learning, sticking in the same predicament as existing MLLMs.**
>
> A1: Thank you for your insightful comments and the opportunity to clarify some aspects of our work. We would like to clarify a potential misunderstanding regarding Parrot's reliance on balanced datasets.
>
> Firstly, constructing a balanced dataset is inherently challenging, especially when dealing with multilingual data. For example, English corpora are significantly larger and more readily available compared to other languages, leading to inherent imbalances in the data. **However, contrary to your impression, our dataset is not balanced.** As shown in **Appendix Table 5** and the additional table below, multilingual data constitutes only a small proportion (~5%) of the entire dataset. This demonstrates that Parrot has been designed and evaluated in an imbalanced data scenario, which reflects real-world situations where imbalanced datasets are common.
>
> Therefore, we specifically design the MoE-based visual token alignment method that aims to address the challenges posed by such imbalanced scenarios. Our approach is intended to improve visual token alignment in multilingual settings, even when the distribution of languages is skewed. Our method does not assume balanced data but instead leverages the inherent structure of MoE to adaptively allocate capacity to different languages, mitigating sub-optimal learning due to data imbalance.
>
> Training Stage|Datasets|Samples|Total|
> |-|-|-|-|
> | **Stage 1**| LLAVA-1.5-pretrain   | 558K| 1.2M  |
> || Laion-Caption | 12K |   |
> || CC12M-Caption  | 645K|   |
> | **Stage 2**| LLAVA-1.5-finetune  | 665K| 793K  |
> || ShareGPT4V-zh | 71K |   |
> || ShareGPT4V-pt   | 14K |   |
> || ShareGPT4V-ar | 12K |   |
> || ShareGPT4V-tr| 17K |   |
> || ShareGPT4V-ru| 14K |   |
>
> On the other hand, to further investigate the scaling law in multilingual settings, we have conducted experiments where we progressively expanded the multilingual data (excluding Chinese and English) until it reached a volume comparable to the amount of Chinese data (~70K). The results, shown in the table below, demonstrate that Parrot still satisfies the multilingual scaling law. For instance, the performance on Portuguese improved by 3.0 points, and Arabic saw a gain of 5.2 points. As we increase the multilingual data, the model's performance on the MMMB benchmark continues to improve, suggesting that our model can handle imbalanced multilingual data while still achieving effective scaling and performance gains.
>
> |Sample Size (each language)|MMMB_en|MMMB_zh|MMMB_pt|MMMB_ar|MMMB_tr|MMMB_ru|
> |-|-|-|-|-|-|-|
> |10K|70.0|68.1|67.3|62.7|58.0|66.3|
> |30K|70.1|68.0|67.6|64.1|59.9|66.7|
> |50K|69.9|67.9|67.8|64.8|61.4|67.2|
> |70K|70.3|68.4|68.3|65.7|63.2|67.4|

---

> ### Author Response · Authors · 2024-11-19
> **Thank you for your detailed, positive, and encouraging review (2/3)**
>
> > **Q2: Lack of a strong baseline. I wonder about the performance of a naive baseline where we first translate the question into English and then translate the English answer back to the target language.**
>
> A2: Thank you for your question regarding the naive baseline of translation. Our experimental setting follows recent work in multilingual and multimodal large language models [1-3], where such a naive baseline has not been commonly considered. And we agree that a translation-based approach could be a straightforward alternative. However, it faces some significant challenges.
>
> First, it is highly susceptible to translation noise, particularly issues related to polysemy and meaning ambiguity between languages. Moreover, our benchmark includes a substantial number of cultural-specific questions, which require deep cultural context knowledge that translation alone cannot effectively capture. In practical use, adding an additional translation step would also introduce extra overhead, increasing both the time and computational cost.
>
> **Despite these challenges, we acknowledge the importance of evaluating this baseline and conducting experiments to assess the performance of this translation-based baseline by using the Google Translation API.** As shown in the table below, the results reveal a "seesaw effect"—while the naive baseline shows some improvements in certain languages, such as Chinese, it leads to performance degradation in others, such as Russian and Portuguese. This highlights the difficulty of addressing multilingualism and multimodal tasks solely through translation.
>
> **We have expanded upon this analysis in Section E.2 of the updated version to further clarify these points.** We hope this provides a clearer perspective on the limitations of translation-based approaches in handling multimodal multilingual tasks.
>
> [1] Large multilingual models pivot zero-shot multimodal learning across languages. ICLR2024
> [2] Respond in my Language: Mitigating Language Inconsistency in Response Generation based on Large Language Models. ACL2024
> [3] Why do LLaVA Vision-Language Models Reply to Images in English? EMNLP2024
>
> |Methods|MMMB_en|MMMB_zh|MMMB_pt|MMMB_ar|MMMB_tr|MMMB_ru|
> |-|-|-|-|-|-|-|
> |LLaVA|67.1|58.8|59.8|43.5|46.4|59.1|
> |LLaVA w/ translation|67.1|60.7|58.6|47.3|48.6|58.9|
> |Parrot|70.0|68.1|67.3|62.7|58.0|66.3|
>
> > **Q3: At the first pretraining stage, is the MoE initialized with random parameters? If yes, how can we learn a good Projector under a random MoE?**
>
> A3: Thank you for your thorough review and valuable feedback on our work. We respond to the concerns below:
>
> During the first pre-training stage, the MoE module is not activated or included in the training process. Instead, we focus exclusively on training the projector. This avoids the issue of training a good projector under a randomly initialized MoE.
>
> In detail:
> 1. **Pre-training Stage:** In this stage, the MoE module is bypassed entirely, meaning the image tokens do not pass through the MoE. Instead, the primary goal of this stage is to train the projector using a large number of image-text pairs. This enables the projector to align image tokens and textual tokens effectively without interference from the untrained MoE module.
> 2. **SFT Stage:** Since the SFT stage requires the participation of MoE modules, we randomly initialize the parameters of the MoE components prior to the SFT phase. Once the projector has been trained and achieves robust alignment capabilities in the pre-training stage, we introduce multilingual training data and activate the MoE parameters. At this stage, the MoE is optimized with textual guidance, which drives the alignment of visual tokens while leveraging the well-trained projector. The prior alignment achieved in the pre-training stage allows the MoE to optimize efficiently during this phase.
>
> **We present the entire training process of parrot in the form of pseudocode, as shown in Algorithm 1 in the appendix**. It is clear from the algorithm that during the pre-training phase, only the projector is trained. Before the start of the SFT phase, the MoE modules are randomly initialized and incorporated into the training process during the SFT phase.
>
> **In this revision, we have supplied further clarification about the MoE training strategy in Section E.1.** This clarification will help to highlight how the MoE module interacts with the projector at each stage and how this contributes to the model’s overall multilingual proficiency.

---

> ### Author Response · Authors · 2024-11-19
> **Thank you for your detailed, positive, and encouraging review (3/3)**
>
> > **Q4: An open question: What is the most real benefit for today's MLLMs? Recent MLLMs can be divided into two groups: 1) dataset-driven, these models adopt simple adaptor to map images into the language space (Qwen-vl, llava, gpt-4o, gemini Pro) and jointly train MLLMs on massive image-text data. 2) tokenization-based models, these models believe a good image tokenization can align the image and text well (Parrot, [1][2]). In my opinion, the simple structure and target datasets fine-tuning may have better robustness and improvement than small structure-based modifications.**
>
> A4: Thank you for your valuable question regarding the benefits of MLLMs in the current landscape. We agree that recent MLLMs can be broadly categorized into two groups: dataset-driven models and tokenization-based models, and **both have their respective strengths and limitations depending on the specific use case.**
>
> Dataset-driven models, such as Qwen-VL, LLaVA, GPT-4o, and Gemini Pro, rely on massive image-text datasets for training and adopt simple adaptors to map images into the language space. These models are generally more robust due to their ability to generalize well across a wide variety of multimodal tasks. The use of extensive training data allows them to handle diverse and complex tasks, but they come with a significant cost in terms of computational resources. Moreover, their performance is heavily dependent on the quality and diversity of the training data, which may pose challenges in specialized domains where relevant data is sparse.
>
> On the other hand, tokenization-based models like Parrot focus on image tokenization to facilitate better alignment between images and text. These models excel in resource-constrained environments, where data or computational resources may be limited. Their specialized architecture allows them to perform efficiently on specific tasks, making them ideal for scenarios where targeted fine-tuning is possible. However, due to the more restricted training data and computational resource constraints, they may struggle with generalization as effectively across diverse datasets compared to dataset-driven models.
>
> In our work, we found that tokenization-based approaches are particularly well-suited for the multilingual tasks we address, where data availability is often limited or imbalanced across languages. This approach allows us to achieve strong alignment between image and text representations without relying on massive datasets. Moreover, we emphasize that **our proposed method is complementary to dataset-driven approaches**. When sufficient data is available, our tokenization-based strategy can be integrated with dataset-driven models to enhance their performance further, combining the strengths of both methodologies.
>
> Additionally, we refer to the results in **Table 6 and Table 7 of the appendix**, where we observe that an increase in the volume of data does not necessarily lead to superior multilingual performance. Under the same model size, our model achieves 87.7 points on the Chinese-English LLaVA-Bench, while Qwen-VL, despite utilizing 100x more data than us, surpasses our score by only 0.5 points. Specifically, models like VisCPM, mPLUG-Owl, and Qwen-VL have relied on extensive Chinese and English datasets (100M+), whereas our model uses less than 2M data points. Despite this disparity, these models do not demonstrate significant advantages over the Parrot model, which benefits from the meticulously designed architecture and limited but carefully curated multilingual data. This highlights that while data quantity is important, data quality and thoughtful architectural design are equally critical for achieving strong multilingual capabilities.
>
> In summary, dataset-driven models offer scalability and robustness, making them suitable for a wide range of multimodal tasks. Tokenization-based models, on the other hand, are more efficient and effective for specific tasks, particularly when resources are limited or when fine-tuning is focused on specific domains. We believe both types of models have their place in the current landscape, depending on the available resources and the problem at hand.

---

> ### Author Response · Authors · 2024-11-25
> **Appreciating Your Reviews and Humbly Asking for Feedback**
>
> Dear Reviewer foLk,
>
> We sincerely appreciate your great efforts in reviewing this paper.
>
> During the remaining hours of the author-reviewer discussion period, it would be great if you could inform us whether our response has addressed your concerns regarding our paper. Your dedication to reviewing our work despite your busy schedule is genuinely appreciated. Lastly, we just want to say thank you for your evaluation of both our paper and our rebuttal.
>
> Best regards,
>
> Authors of paper 5775

---

> ### Comment · Reviewer_foLk · 2024-11-26
>
> I thank the authors for their detailed response. As most of my concerns have been addressed, I decided to raise my score to 7.

---

> > ### Author Response · Authors · 2024-11-26
> > **Many thanks!**
> >
> > Thank you very much for your positive feedback and for your willingness to consider raising the score. We are pleased that we were able to address your concerns and are open to continuing the discussion should you have any further questions or concerns.

---

> ### Author Response · Authors · 2024-11-28
> **A friendly reminder**
>
> Dear Reviewer foLk,
>
> Thank you so much for your thoughtful and insightful feedback. We truly appreciate your time, effort, and support in raising the score. Just a friendly reminder that the updated score hasn’t been reflected in the system yet. Wishing you a happy Thanksgiving!
>
> With warm regards,
>
> Authors of paper 5775

---

### Author Response · Authors · 2024-11-19
**General Response**

We would like to express our deepest gratitude to the reviewers for the meticulous examination of the paper and their insightful and valuable comments. We acknowledge that all the reviewers observed the shining point, saying our work is **well-written, interesting, easy to follow, and comprehensive** (foLK, F1M5, cBkD, iiKz). They also consider our proposed method as well as the new multilingual benchmark meaningful (foLK, F1M5, cBkD), shedding light on the MLLM community (cBkD), and being useful for subsequent research (foLK). They agree on the insightful observation of the lack of balance among different languages, too (foLK, cBkD, iiKz). Additionally, all the reviewers acknowledge that extensive experiments validate the improved performance of our proposed method (foLK, F1M5, cBkD, iiKz).

In this rebuttal, we have given careful thought to the reviewers’ suggestions and made the following revisions to our manuscript to answer the questions and concerns:

- In **Supplementary Section D.6**, we add numerical results about the multilingual data scaling and model size scaling;
- In **Supplementary Section D.7**, we add experiments about the baseline LLaVA using the same multilingual data as Parrot;
- In **Supplementary Section D.8**, we add the experiments to compare Parrot with Qwen2-VL and LLaVA-OV and extend Parrot with Qwen2-7B LLM;
- In **Supplementary Section E.1**, we add a more detailed description and pseudocode of the MoE training strategy;
- In **Supplementary Section E.2**, we add the experiments about the translation-based baseline and discussions about the challenges when using this approach;
- In **Supplementary Section E.3**, we add the description about the construction of our in-house dataset;
- We have uploaded the source code of Parrot as the **Supplementary Material**;
- We have uploaded the training dataset of Parrot to an anonymous GitHub repository (**[Code and Dataset](https://anonymous.4open.science/r/Parrot-Anonymous-FDC2)**).

We have highlighted the revised part in our manuscript in **blue** color. Please check the answers to specific comments.

---

### Author Response · Authors · 2024-11-22
**Appreciating Your Reviews and Humbly Ask for Feedback**

Dear Reviewers,

During the remaining time of the author-reviewer discussion period, it would be great if you could inform us whether our response has addressed your concerns regarding our paper. Your dedication to reviewing our work despite your busy schedule is genuinely appreciated. Lastly, we just want to say thank you for your evaluation of both our paper and our rebuttal.

Kind regards,

Authors

---

### Meta-Review · Area_Chair_rLzh · 2024-12-18

**Metareview:**

(a) Summary:
The paper introduces Parrot, a Mixture-of-Experts (MoE)-based model to improve multilingual alignment in Multimodal Large Language Models (MLLMs). It addresses imbalances in non-English data and proposes MMMB, a multilingual benchmark. Empirical results demonstrate Parrot’s improved performance across benchmarks compared to baselines.

(b) Strengths:
Introduction of a new evaluation benchmark (MMMB).

(c) Weaknesses:
1) Baseline comparisons lack rigor; fair comparisons across pre-trained models and hyperparameters are missing.
2) The MoE's effectiveness in low-resource scenarios is not fully resolved.

(d) Decision:
Reject. While the paper presents a well-motivated study, the limited novelty, incremental technical contributions, and incomplete baseline comparisons lead to rejection.

**Additional Comments On Reviewer Discussion:**

The rebuttal clarified the model’s MoE training strategy, addressed data imbalance concerns, and added experiments for stronger baselines. However, reviewers noted limited novelty, incremental contributions, and unresolved fair comparisons. Despite improvements, concerns persisted, leading to a decision to reject based on these critical limitations.

---

### Decision · Program_Chairs · 2025-01-22

Reject